# GACL: Exemplar-Free Generalized Analytic Continual Learning

**Huiping Zhuang**[1]* **Yizhu Chen**[1]* **Di Fang**[1] **Run He**[1] **Kai Tong**[1]
**Hongxin Wei**[2] **Ziqian Zeng**[1†] **Cen Chen**[1,3,4†]
[1]South China University of Technology, China
[2]Southern University of Science and Technology, China
[3]Shenzhen Institute, Hunan University, China
[4]Pazhou Lab, China

## Abstract

Class incremental learning (CIL) trains a network on sequential tasks with separated categories in each task but suffers from catastrophic forgetting, where models quickly lose previously learned knowledge when acquiring new tasks. The generalized CIL (GCIL) aims to address the CIL problem in a more real-world scenario, where incoming data have mixed data categories and unknown sample size distribution. Existing attempts for the GCIL either have poor performance or invade data privacy by saving exemplars. In this paper, we propose a new exemplar-free GCIL technique named generalized analytic continual learning (GACL). The GACL adopts analytic learning (a gradient-free training technique) and delivers an analytical (i.e., closed-form) solution to the GCIL scenario. This solution is derived via decomposing the incoming data into exposed and unexposed classes, thereby attaining a weight-invariant property, a rare yet valuable property supporting an equivalence between incremental learning and its joint training. Such an equivalence is crucial in GCIL settings as data distributions among different tasks no longer pose challenges to adopting our GACL. Theoretically, this equivalence property is validated through matrix analysis tools. Empirically, we conduct extensive experiments where, compared with existing GCIL methods, our GACL exhibits a consistently leading performance across various datasets and GCIL settings. Source code is available at `https://github.com/CHEN-YIZHU/GACL`.

## 1 Introduction

Class incremental learning (CIL) [1], an important form of continual learning, aims to effectively tune an off-the-shelf network on incoming new datasets, with data excluding various categories from its previous states. The CIL has gained significant traction due to its ability to refine learned models for new and unfamiliar data classes, eliminating the need to start the training process from scratch. This elimination of retraining saves valuable computational resources, which is especially important in the era of pre-trained models that have absorbed a massive amount of data.

One significant challenge in CIL is *catastrophic forgetting* [2, 3], which causes trained models to lose existing knowledge when gaining new information quickly. This can be attributed to the fundamental property of gradient-based iterative algorithms that impose a *task-recency* bias, i.e., predictions favor recently updated categories [4]. To the authors' knowledge, no solutions exist for these gradient-trained CIL models to fully tackle catastrophic forgetting.

---

*These authors contribute equally.
†Corresponding authors: Ziqian Zeng (zqzeng@scut.edu.cn) and Cen Chen (chencen@scut.edu.cn).

38th Conference on Neural Information Processing Systems (NeurIPS 2024).

On the other hand, traditional CIL assumes that the number of samples in each task is fixed and that new tasks are entirely disjoint from previous ones. This paradigm does not align with real-world scenarios, where training data may include both new and previously encountered categories, and the number of data points often exhibits arbitrariness in each task. This extended CIL setting is referred to as generalized CIL (GCIL) [5, 6]. Such an uneven task-wise distribution of training samples and data categories further complicates the forgetting issue. For instance, GCIL may lead to the neglect of minority samples within a batch, thereby undermining representation during the training process.

To mitigate catastrophic forgetting, a simple but effective approach is to replay historical samples. Replay-based CIL [1, 4] mitigates forgetting by storing a small number of samples from historical categories for the model to review while learning new information. However, this replay mechanism poses risks to data privacy. Thus, the exemplar-free CIL (EFCIL) without saving old exemplars gains prominence due to the increasing concern for privacy. However, many EFCIL methods perform poorly due to the task-recency bias caused by the nature of gradient-based algorithms [4]. Recently, this dilemma has been alleviated by the analytic continual learning (ACL) [7, 8], an emerging EFCIL branch that first achieves comparable or even more competitive performance over the replay-based CIL. This improvement occurs because, for the first time, ACL achieves a near "complete non-forgetting" by allowing an equivalence between the incremental learning and its joint training (i.e., the weight-invariant property).

The ACL provides a powerful toolbox for traditional EFCIL scenarios where data categories among training tasks are mutually exclusive. However, an apparent gap exists between the existing ACL techniques and the more desired and real-world GCIL scenario. Exploring the possibility of incorporating the weight-invariant property into the GCIL framework is both a significant and natural motivation, as it has the potential to enhance overall performance. To achieve this, we propose a generalized analytic continual learning (GACL), a new and compensated ACL member, offering a weight-invariant property solution to the GCIL. The key contributions are summarized as follows.

- We present the GACL, an exemplar-free technique that achieves the equivalence between the GCIL (with split incoming data) and its joint training (with data centralized in a single task).
- We theoretically establish the GACL's weight-invariant property. It is achieved and proved by separating the incoming data into exposed and unexposed components and aligning them structurally with matrix decomposition techniques.
- We isolate the distinctive component of the GACL, namely the *exposed class label gain* (ECLG), from the existing ACL. This module explains the feasibility of achieving GCIL's analytic learning, offering a high interpretability in the GCIL realm.
- Experiments on various benchmark datasets are presented, showing that the GACL outperforms the existing EFCIL by a large margin. It also exceeds most state-of-the-art replay-based methods.

## 2 Related Works

This section reviews existing methods for CIL and its more real-world counterpart, i.e., GCIL.

### 2.1 CIL Techniques

Existing CIL methods can be roughly divided into three categories: replay-based methods, regularization-based methods, and prototype-based methods.

The *replay-based CIL* methods such as iCaRL [1], LUCIR [4], PODNet [9], AANets [10], FOSTER [11], and OHO [12], retain past training samples as exemplars and utilize them during the learning of new ones. However, storing original training samples presents a significant challenge, particularly in scenarios with strict data privacy requirements.

The *regularization-based CIL* aims to design a loss function that prevents the change of activations or important weights. Methods such as the Less-forgetting learning [13] and the LwF [14] introduce knowledge distillation [15] into their loss function to prevent the forgetting caused by activation drift. EWC [16], EWC++, RWalk [17], and Rotate your Networks [18], introduce regularization that slows down learning on the weights important for old tasks by calculating the Fisher information matrix.

The *prototype-based CIL* maintains distinct prototypes for each category, which prevents overlapping representations of new and old categories. For example, the PASS [19] distinguishes prior categories

by augmenting feature prototypes. The SSRE technique [20] enhances the dissimilarity between old and new categories via selecting prototypes to incorporate with new samples into a distillation process. The FeTrIL [21] uses new representations to generate pseudo-features of old categories.

## 2.2 Analytic Continual Learning

The ACL is a recently developed EFCIL branch inspired by the analytic learning [22, 23, 24] where the training of neural networks yields a closed-form solution using least squares. The ACIL [7] first converts a continual learning problem to a batch recursive least-squares problem, eliminating the need to store samples by preserving the correlation matrix, and the RanPAC [25] applies this trick to pre-trained models. The GKEAL [8] focuses on the few-shot CIL scenarios by leveraging a Gaussian kernel projection. The DS-AL [26] introduces an additional linear classifier to learn the residue of the ACIL to enhance the plasticity, while the REAL [27] introduces the representation enhancing distillation to improve the backbone's generalization capabilities. The AFL [28] extends the ACL to federated learning, transitioning from temporal increment to spatial increment, and similar techniques are applied to the reinforcement learning [29]. The ACL is an emerging competitive CIL branch with a closed-form solution that leads to a valuable weight-invariant property, securing the equivalence between CIL and its joint learning. However, existing ACL methods are designed for the CIL scenario in which the categories of samples in each task must be entirely distinct. This restricts their applicability in real-world scenarios.

## 2.3 The Generalized Class Incremental Learning

The GCIL simulates real-world incremental learning, as distributions of data category and size could be unknown in one task. The GCIL arouses problems such as intra- and inter-task forgettings and the class imbalance problem [30]. The key GCIL properties can be summarized as follows: (i) the number of classes across different tasks is not fixed; (ii) classes shown in prior tasks could reappear in later tasks; (iii) training samples are imbalanced across different classes in each task [6] (See Appendix B).

There are several GCIL settings. In the BlurryM setting [5], $a\%$ of the classes are disjoint between tasks, while the remaining classes appear in every task. The i-Blurry-N-M [31] setting has blurry task boundaries and requires the model to perform anytime inference. However, the i-Blurry scenario has a fixed number of classes in each task with the same proportion of new and old classes. The Si-Blurry [30] is the most complex and realistic GCIL setting satisfying all three GCIL properties since it has an ever-changing number of classes and is capable of effectively simulating newly emerging or disappearing data, highlighting the problem of uneven distribution in real-world scenarios.

To address the issue of the GCIL, gradient-based sample selection methods such as the GSS-IQP and the GSS-Greedy are proposed by [5]. The RM [32] proposes a memory management strategy based on per-sample classification uncertainty and data augmentation, while the management in the CLIB [31] eliminates samples based on a per-sample importance score. The DualPrompt [33], as an EFCIL method, introduces the prompt-based learning to the CIL problem, and the MVP [30] proposes an instance-wise logit masking and contrastive visual prompt tuning loss.

## 3 The Proposed Method

In this section, we deliver details of the proposed GACL. We first define the learning problem. Then, we derive the GACL by employing matrix decomposition techniques. A corresponding theoretical analysis follows to indicate the interpretability of our work. An overview is depicted in Figure 1.

## 3.1 Problem Definition

We denote the complete set of available data as $\mathcal{D}$. When $\mathcal{D}$ is partitioned into a sequence of GCIL tasks, we assume that $\mathcal{D}_k^{\text{train}} \sim \{\boldsymbol{X}_k^{\text{train}}, \boldsymbol{Y}_k^{\text{train}}\}$ is the set of training samples that are present in task $k$. The training dataset $\mathcal{D}_k^{\text{train}}$ consists of labeled samples, where $\boldsymbol{X}_k^{\text{train}} \in \mathbb{R}^{N_k \times c \times w \times h}$ represents $N_k$ input image samples with a shape of $c \times w \times h$. $\boldsymbol{Y}_k^{\text{train}} \in \mathbb{R}^{N_k \times d_{y_k}}$ represents $N_k$-stacked one-hot encoded label tensors with $d_{y_k}$ classes that have been seen from task 1 to task $k$. $\mathcal{D}_k^{\text{test}} \sim \{\boldsymbol{X}_k^{\text{test}}, \boldsymbol{Y}_k^{\text{test}}\}$

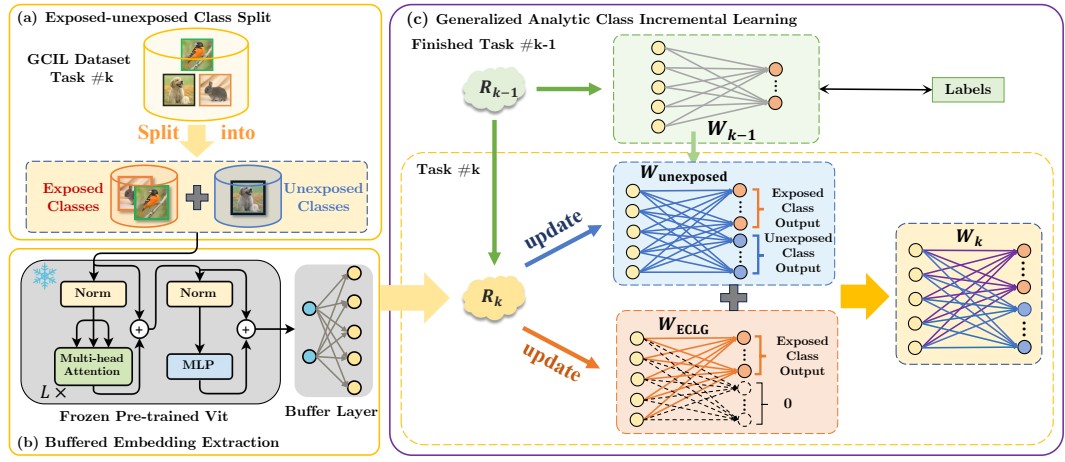

Figure 1: An overview of our proposed GACL. (a) Labels of the *exposed class* and the *unexposed class* are extracted in each GCIL task (see definition in Section 3.2), respectively. (b) A frozen pre-trained ViT and a buffer layer are utilized to extract features from the inputs. (c) The key to the recursively updated formulation of the GACL contains two components. The $\hat{\boldsymbol{W}}_{\text{unexposed}}^{(k)}$ takes in the contribution of unexposed class data (see (11)). The other is contributed by the ECLG module $\hat{\boldsymbol{W}}_{\text{ECLG}}^{(k)}$ (e.g., see (12)), which reflects the gain of exposed class data on the seen categories. The recursive formulation flows aided by the *autocorrelation memory matrix* $\boldsymbol{R}$ throughout the GCIL.

is the test dataset in task $k$. The goal of GCIL in task $k$ is to train networks using $\mathcal{D}_k^{\text{train}}$ and evaluate their performance on the test dataset $\mathcal{D}_{1:k}^{\text{test}}$. Here, $\mathcal{D}_{1:k}$ denotes the joint dataset spanning tasks 1 to $k$.

### 3.2 Exposed-unexposed Class Split

In each GCIL task, classes may not appear exclusively. Hence, in any GCIL task $k$, we refer to classes that have appeared in previous tasks 1 to $k-1$ as the *exposed classes* of task $k$, while classes making their initial appearance are the *unexposed classes* of task $k$ as shown in Figure 1 (a). This distinction helps to characterize the evolving nature of class occurrences throughout different GCIL tasks.

In a task-wise GCIL scenario, we can involve all class labels in a set $\mathcal{S}$. In task $k$, the set of the exposed class labels is denoted as $\mathcal{S}_{\text{exposed, k}} \subseteq \mathcal{S}$, while the set of unexposed class labels is marked by $\mathcal{S}_{\text{unexposed, k}} \subseteq \mathcal{S}$, where $\mathcal{S}_{\text{exposed, k}} \cap \mathcal{S}_{\text{unexposed, k}} = \varnothing$. Note that $\mathcal{S}_{\text{exposed, k}}$ and $\mathcal{S}_{\text{unexposed, k}}$ may evolve from task $k-1$ to task $k$, that is

$$\mathcal{S}_{\text{exposed, k}} = S_{\text{unexposed, k-1}} \cup \mathcal{S}_{\text{exposed, k-1}} = S_{\text{unexposed, k-1}} \cup S_{\text{unexposed, k-2}} \ldots \cup \mathcal{S}_{\text{unexposed, 1}}.$$

From the scope of exposed-unexposed classes, the $d_{y_k}$ can be represented as $d_{y_k} = |\mathcal{S}_{\text{exposed, k}}| + |\mathcal{S}_{\text{unexposed, k}}| = d_{y_{k-1}} + |\mathcal{S}_{\text{unexposed, k}}|$, where $|\cdot|$ denotes the cardinality of a set.

In task $k$, given training dataset $\mathcal{D}_k^{\text{train}} \sim \{\boldsymbol{X}_k^{\text{train}}, \boldsymbol{Y}_k^{\text{train}}\}$, class labels $\boldsymbol{Y}_k^{\text{train}}$ can be partitioned due to the *exposed-unexposed split* as follows:

$$\boldsymbol{Y}_k^{\text{train}} = \begin{bmatrix} \bar{\boldsymbol{Y}}_k^{\text{train}} & \tilde{\boldsymbol{Y}}_k^{\text{train}} \end{bmatrix}, \tag{1}$$

where $\bar{\boldsymbol{Y}}_k^{\text{train}} \in \mathbb{R}^{N_k \times d_{y_{k-1}}}$ is the *exposed class label matrix* and $\tilde{\boldsymbol{Y}}_k^{\text{train}} \in \mathbb{R}^{N_k \times (d_{y_k} - d_{y_{k-1}})}$ is the *unexposed class label matrix*. They correspond to segments displaying the appearance of exposed classes and unexposed classes.

### 3.3 Buffered Embedding Extraction

The power of pre-trained models allows the GACL to adopt a frozen backbone from structures such as ViT [34] to extract the features of images shown in Figure 1 (b). Let

$$\boldsymbol{X}^{(\text{E})} = f_{\text{backbone}}(\boldsymbol{X}, \boldsymbol{\Theta}_{\text{backbone}}) \tag{2}$$

be the features extracted by the backbone, where $\mathbf{\Theta}_{\text{backbone}}$ indicates the backbone weight. Then, we use a buffer layer to project features, i.e.,

$$\boldsymbol{X}_i^{(\text{B})} = f_{\text{buffer}}(\boldsymbol{X}^{(\text{E})}), \tag{3}$$

where $f_{\text{buffer}}$ indicates the operation of the buffer layer. Several options for the buffer layer exist, including a randomly initialized linear mapping in the ACIL [7] or a kernel embedding projection in the GKEAL [8]. The selection of the buffer layer is not our focus. For convenience, we follow the ACIL, taking the random linear projection followed by a non-linear activation function as the buffer layer, i.e. $f_{\text{buffer}}(\boldsymbol{X}^{(\text{E})}) = \text{ReLU}(\boldsymbol{X}^{(\text{E})}\boldsymbol{W}_{\text{B}})$, where the elements of the buffer layer weight $\boldsymbol{W}_{\text{B}}$ are randomly sampled from a normal distribution.

## 3.4 Generalized Analytic Class Incremental Learning

Here, we derive the GACL by partitioning training samples into unexposed and exposed categories, as shown in Figure 1 (c). Let $\boldsymbol{X}_{1:k}^{\text{total}}$ and $\boldsymbol{Y}_{1:k}^{\text{total}}$ be the accumulated feature and label matrices in task $k$, which can be extended from the accumulated matrices $\boldsymbol{X}_{1:k-1}^{\text{total}}$ and $\boldsymbol{Y}_{1:k-1}^{\text{total}}$ in task $k-1$ as follows.

$$\boldsymbol{X}_{1:k}^{\text{total}} = \begin{bmatrix} \boldsymbol{X}_{1:k-1}^{\text{total}} \\ \boldsymbol{X}_k^{(\text{B})} \end{bmatrix}, \quad \boldsymbol{Y}_{1:k}^{\text{total}} = \begin{bmatrix} \boldsymbol{Y}_{1:k-1}^{\text{total}} & \boldsymbol{0} \\ \bar{\boldsymbol{Y}}_k^{\text{train}} & \tilde{\boldsymbol{Y}}_k^{\text{train}} \end{bmatrix}.$$

Subsequently, one could formulate the learning problem in task $k$ by a fully connected network (FCN) as the classifier

$$\underset{\boldsymbol{W}_{\text{FCN}}^{(k)}}{\arg\min} \ \left\| \boldsymbol{Y}_{1:k}^{\text{total}} - \boldsymbol{X}_{1:k}^{\text{total}} \boldsymbol{W}_{\text{FCN}}^{(k)} \right\|_{\text{F}}^2 + \gamma \left\| \boldsymbol{W}_{\text{FCN}}^{(k)} \right\|_{\text{F}}^2, \tag{4}$$

where $\|\cdot\|_{\text{F}}$ is Frobenius-norm, $\gamma \geq 0$ is the regularization term and $\boldsymbol{W}_{\text{FCN}}^{(k)}$ indicates the FCN layer weight. The optimal solution to (4) is

$$\hat{\boldsymbol{W}}_{\text{FCN}}^{(k)} = (\boldsymbol{X}_{1:k}^{\text{total}\top} \boldsymbol{X}_{1:k}^{\text{total}} + \gamma \boldsymbol{I})^{-1} \boldsymbol{X}_{1:k}^{\text{total}\top} \boldsymbol{Y}_{1:k}^{\text{total}}. \tag{5}$$

The goal of the GACL is then to obtain $\hat{\boldsymbol{W}}_{\text{FCN}}^{(k)}$ recursively from $\hat{\boldsymbol{W}}_{\text{FCN}}^{(k-1)}$ without directly involving historical samples (e.g., $\boldsymbol{X}_{1:k-1}^{\text{total}}$ and $\boldsymbol{Y}_{1:k-1}^{\text{total}}$). That is to solve

$$\underset{\boldsymbol{W}_{\text{FCN}}^{(k)}}{\arg\min} \ \left\| \begin{bmatrix} \boldsymbol{Y}_{1:k-1}^{\text{total}} & \boldsymbol{0} \\ \bar{\boldsymbol{Y}}_k^{\text{train}} & \tilde{\boldsymbol{Y}}_k^{\text{train}} \end{bmatrix} - \begin{bmatrix} \boldsymbol{X}_{1:k-1}^{\text{total}} \\ \boldsymbol{X}_k^{(\text{B})} \end{bmatrix} \boldsymbol{W}_{\text{FCN}}^{(k)} \right\|_{\text{F}}^2 + \gamma \left\| \boldsymbol{W}_{\text{FCN}}^{(k)} \right\|_{\text{F}}^2 \tag{6}$$

by recursively updating the previous-task weight $\hat{\boldsymbol{W}}_{\text{FCN}}^{(k)}$. To achieve this, we define an *autocorrelation memory matrix* as follows.

$$\boldsymbol{R}_k = (\boldsymbol{X}_{1:k}^{\text{total}\top} \boldsymbol{X}_{1:k}^{\text{total}} + \gamma \boldsymbol{I})^{-1}. \tag{7}$$

Accordingly, we summarize the recursive formulation of the proposed GACL in Theorem 3.1.

**Theorem 3.1.** *Let $\hat{\boldsymbol{W}}_{\text{FCN}}^{(k)}$ be the optimal estimation of* (6) *with all the training data from task 1 to task $k$. Then $\hat{\boldsymbol{W}}_{\text{FCN}}^{(k)}$ is equivalent to its recursive form*

$$\hat{\boldsymbol{W}}_{\text{FCN}}^{(k)} = \left[ \hat{\boldsymbol{W}}_{\text{FCN}}^{(k-1)} - \boldsymbol{R}_k \boldsymbol{X}_k^{(\text{B})\top} \boldsymbol{X}_k^{(\text{B})} \hat{\boldsymbol{W}}_{\text{FCN}}^{(k-1)} + \boldsymbol{R}_k \boldsymbol{X}_k^{(\text{B})\top} \bar{\boldsymbol{Y}}_k^{\text{train}} \quad \boldsymbol{R}_k \boldsymbol{X}_k^{(\text{B})\top} \tilde{\boldsymbol{Y}}_k^{\text{train}} \right], \tag{8}$$

*where*

$$\boldsymbol{R}_k = \boldsymbol{R}_{k-1} - \boldsymbol{R}_{k-1} \boldsymbol{X}_k^{(\text{B})\top} (\boldsymbol{I} + \boldsymbol{X}_k^{(\text{B})} \boldsymbol{R}_{k-1} \boldsymbol{X}_k^{(\text{B})\top})^{-1} \boldsymbol{X}_k^{(\text{B})} \boldsymbol{R}_{k-1}. \tag{9}$$

*Proof.* See Appendix A. □

As indicated in Theorem 3.1, the weight $\hat{\boldsymbol{W}}_{\text{FCN}}^{(k)}$ in task $k$ recursively obtained using the previous-task weight $\hat{\boldsymbol{W}}_{\text{FCN}}^{(k-1)}$ is identical to its joint-learning counterpart formulated in (6). That is, the GACL maintains the same *weight-invariant property* in the GCIL scenario as other ACL methods.

**Algorithm 1** The pseudo-code of GACL.

---

**Input:** GCIL tasks $\mathcal{D}_1^{\text{train}}, \ldots, \mathcal{D}_K^{\text{train}}$ with $\mathcal{D}_k^{\text{train}} \sim \{X_k^{\text{train}}, Y_k^{\text{train}}\}$, the pre-trained backbone with frozen weight $\Theta_{\text{backbone}}$

**Initialization:** $R_0 \leftarrow \gamma I$, $W_{\text{FCN}}^{(0)} \leftarrow 0$

**for** task $k = 1$ **to** $K$ **do**

$\quad X_k^{(\text{E})} \leftarrow f_{\text{backbone}}(X_k^{\text{train}}, \Theta_{\text{backbone}})$ (2)

$\quad X_k^{(\text{B})} \leftarrow f_{\text{buffer}}(X_k^{(\text{E})})$ (3)

$\quad$ Decompose $Y_k^{\text{train}}$ into exposed and unexposed class components $\bar{Y}_k^{\text{train}}$ and $\tilde{Y}_k^{\text{train}}$

$\quad R_k \leftarrow R_{k-1} - R_{k-1}X_k^{(\text{B})\top}(I + X_k^{(\text{B})}R_{k-1}X_k^{(\text{B})\top})^{-1}X_kR_{k-1}$ (9)

$\quad W_{\text{unexposed}}^{(k)} \leftarrow \left[W_{\text{FCN}}^{(k-1)} - R_kX_k^{(\text{B})\top}X_k^{(\text{B})}\hat{W}_{\text{FCN}}^{(k-1)} \quad R_kX_k^{(\text{B})\top}\tilde{Y}_k^{\text{train}}\right]$ (11)

$\quad W_{\text{ECLG}}^{(k)} \leftarrow \left[R_kX_k^{(\text{B})\top}\bar{Y}_k^{\text{train}} \quad 0\right]$ (12)

$\quad W_{\text{FCN}}^{(k)} \leftarrow W_{\text{unexposed}}^{(k)} + W_{\text{ECLG}}^{(k)}$

**end for**

---

The pseudo-code of the GACL is listed in Algorithm 1.

**Exemplar-free.** The recursive formulation is aided by $R_k$ as indicated in (9). Note that this autocorrelation memory matrix records the inverse of inner products among the historical embedding matrices as shown in (7). Hence, the embeddings (e.g., $X_k^{(\text{B})}$) are not reversible. Saving $R_k$ instead of used samples is a safe alternative to preserve past knowledge. That is, our GACL is an *exemplar-free* technique without the need to keep any historical samples.

To more properly explain our GACL, as indicated in Figure 1 (c), the recursive solution in (8) can be rewritten as the sum of the unexposed-class contributed weight $\hat{W}_{\text{unexposed}}^{(k)}$ and the ECLG weight $\hat{W}_{\text{ECLG}}^{(k)}$, i.e.,

$$\hat{W}_{\text{FCN}}^{(k)} = \hat{W}_{\text{unexposed}}^{(k)} + \hat{W}_{\text{ECLG}}^{(k)}, \tag{10}$$

where

$$\hat{W}_{\text{unexposed}}^{(k)} = \left[\hat{W}_{\text{FCN}}^{(k-1)} - R_kX_k^{(\text{B})\top}X_k^{(\text{B})}\hat{W}_{\text{FCN}}^{(k-1)} \quad R_kX_k^{(\text{B})\top}\tilde{Y}_k^{\text{train}}\right], \tag{11}$$

$$\hat{W}_{\text{ECLG}}^{(k)} = \left[R_kX_k^{(\text{B})\top}\bar{Y}_k^{\text{train}} \quad 0\right]. \tag{12}$$

**Unexposed-class Contributed Weight.** The unexposed-class contributed weight $\hat{W}_{\text{unexposed}}^{(k)}$ is recursively updated by the data of the unexposed class only. Note that the unexposed class label $\tilde{Y}_k^{\text{train}}$ is applied on the concatenated weight along with new data $X_k^{(\text{B})\top}$, which is reasonable as historical information should not intervene with the weight update of unseen classes. On the other hand, new data $X_k^{(\text{B})\top}$ could also affect historical knowledge. This is marked by the gain of $-R_kX_k^{(\text{B})\top}X_k^{(\text{B})}\hat{W}_{\text{FCN}}^{(k-1)}$ to the original weight $\hat{W}_{\text{FCN}}^{(k-1)}$ as indicated in (11).

**Exposed-class Label Gain Weight.** The ECLG module indicated in (12) captures knowledge from exposed-class labels. The supervision of this weight component marked by $R_kX_k^{(\text{B})\top}\bar{Y}_k^{\text{train}}$ is mainly contributed by the exposed-class labels (i.e., $\bar{Y}_k^{\text{train}}$). It is important to note that when $\bar{Y}_k^{\text{train}}$ is empty (i.e., no classes reappear in task $k$), this component does not contribute to the update of $\hat{W}_{\text{FCN}}^{(k)}$. This module is also isolated to distinguish GACL's difference from the existing ACL methods in a mathematical analysis manner (indicated as follows).

**Difference from Existing ACL Methods.** Overall, the GACL can be treated as a nontrivial generalization of ACIL [7], GKEAL [8], and various other ACL methods. For instance, in conventional CIL where no classes reappear in new tasks (i.e., $\forall k$, $\bar{Y}_k^{\text{train}} \in \mathbb{R}^{*\times 0}$), the classifier of the GACL $\hat{W}_{\text{FCN}}^{(k)} = \hat{W}_{\text{unexposed}}^{(k)}$, which is equivalent to the recursive classifier of the ACIL. That is, the ACIL is a special case of our proposed GACL. The major difference lies in the ECLG module, corresponding to the exposed-class gain. This pattern makes sense as there must be compensation on top of ACIL updates (specifically designed for traditional CIL) when exposed data (out of setting) participate.

Table 1: Comparison of $\mathcal{A}_{\text{AUC}}$, $\mathcal{A}_{\text{Avg}}$, and $\mathcal{A}_{\text{Last}}$ among the GACL and other methods under the Si-Blurry setting. Data in **bold** represent the best EFCIL results, and data underlined are the best among all settings. We run all experiments 5 times and show "mean$_{\pm\text{standard error}}$".

| Mem Size | Method | EFCIL | CIFAR-100 (%) | | | ImageNet-R (%) | | | Tiny-ImageNet (%) | | |
|---|---|---|---|---|---|---|---|---|---|---|---|
| | | | $\mathcal{A}_{\text{AUC}}$ | $\mathcal{A}_{\text{Avg}}$ | $\mathcal{A}_{\text{Last}}$ | $\mathcal{A}_{\text{AUC}}$ | $\mathcal{A}_{\text{Avg}}$ | $\mathcal{A}_{\text{Last}}$ | $\mathcal{A}_{\text{AUC}}$ | $\mathcal{A}_{\text{Avg}}$ | $\mathcal{A}_{\text{Last}}$ |
| 2000 | EWC++ [16] | ✗ | $53.31_{\pm1.70}$ | $50.95_{\pm1.50}$ | $52.55_{\pm0.71}$ | $36.31_{\pm0.72}$ | $39.87_{\pm1.35}$ | $29.52_{\pm0.43}$ | $52.43_{\pm0.52}$ | $54.61_{\pm1.54}$ | $37.67_{\pm0.77}$ |
| | ER [35] | ✗ | $56.17_{\pm1.84}$ | $53.80_{\pm1.46}$ | $55.60_{\pm0.69}$ | $39.31_{\pm0.70}$ | $43.03_{\pm1.19}$ | $32.09_{\pm0.44}$ | $55.69_{\pm0.47}$ | $57.87_{\pm1.42}$ | $41.10_{\pm0.57}$ |
| | RM [32] | ✗ | $53.22_{\pm1.82}$ | $52.99_{\pm1.69}$ | $55.25_{\pm0.61}$ | $32.34_{\pm1.88}$ | $36.46_{\pm2.23}$ | $25.26_{\pm1.08}$ | $49.28_{\pm0.43}$ | $57.74_{\pm1.57}$ | $41.79_{\pm0.34}$ |
| | MVP-R [30] | ✗ | $60.62_{\pm1.03}$ | $57.58_{\pm0.56}$ | $64.30_{\pm0.29}$ | $47.16_{\pm1.00}$ | $50.36_{\pm0.90}$ | $42.05_{\pm0.15}$ | $61.15_{\pm0.86}$ | $62.41_{\pm0.50}$ | $51.12_{\pm0.67}$ |
| 500 | EWC++ [16] | ✗ | $48.31_{\pm1.81}$ | $44.56_{\pm0.96}$ | $40.52_{\pm0.83}$ | $32.81_{\pm0.76}$ | $35.54_{\pm1.69}$ | $23.43_{\pm0.61}$ | $45.30_{\pm0.61}$ | $46.34_{\pm2.05}$ | $27.05_{\pm1.35}$ |
| | ER [35] | ✗ | $51.59_{\pm1.94}$ | $48.03_{\pm0.80}$ | $44.09_{\pm0.80}$ | $35.96_{\pm0.72}$ | $39.01_{\pm1.54}$ | $26.14_{\pm0.44}$ | $48.95_{\pm0.58}$ | $50.44_{\pm1.71}$ | $29.97_{\pm0.75}$ |
| | RM [32] | ✗ | $41.07_{\pm1.30}$ | $38.10_{\pm0.59}$ | $32.66_{\pm0.34}$ | $22.45_{\pm0.62}$ | $22.08_{\pm1.78}$ | $9.61_{\pm0.13}$ | $36.66_{\pm0.40}$ | $38.83_{\pm2.33}$ | $18.23_{\pm0.22}$ |
| | MVP-R [30] | ✗ | $56.20_{\pm1.47}$ | $53.61_{\pm0.04}$ | $55.35_{\pm0.43}$ | $43.28_{\pm1.41}$ | $45.74_{\pm0.97}$ | $35.60_{\pm1.18}$ | $55.28_{\pm1.42}$ | $55.45_{\pm1.02}$ | $40.12_{\pm0.40}$ |
| 0 | LwF [14] | ✓ | $40.71_{\pm2.13}$ | $38.49_{\pm0.56}$ | $27.03_{\pm2.92}$ | $29.41_{\pm0.83}$ | $31.95_{\pm1.86}$ | $19.67_{\pm1.27}$ | $39.88_{\pm0.90}$ | $41.35_{\pm2.59}$ | $24.93_{\pm2.01}$ |
| | L2P [36] | ✓ | $42.68_{\pm2.70}$ | $39.89_{\pm0.45}$ | $28.59_{\pm3.34}$ | $30.21_{\pm0.91}$ | $32.21_{\pm1.73}$ | $18.01_{\pm3.07}$ | $41.67_{\pm1.17}$ | $42.53_{\pm2.52}$ | $24.78_{\pm2.31}$ |
| | DualPrompt [33] | ✓ | $41.34_{\pm2.59}$ | $38.59_{\pm0.68}$ | $22.74_{\pm3.40}$ | $30.44_{\pm0.88}$ | $32.54_{\pm1.84}$ | $16.07_{\pm3.20}$ | $39.16_{\pm1.13}$ | $39.81_{\pm3.03}$ | $20.42_{\pm3.37}$ |
| | MVP [30] | ✓ | $45.07_{\pm2.43}$ | $44.93_{\pm0.54}$ | $39.94_{\pm0.47}$ | $35.77_{\pm2.55}$ | $35.58_{\pm1.20}$ | $22.06_{\pm5.01}$ | $46.43_{\pm3.07}$ | $45.41_{\pm1.09}$ | $28.21_{\pm2.89}$ |
| | SLDA [37] | ✓ | $53.00_{\pm3.85}$ | $50.09_{\pm2.77}$ | $61.79_{\pm3.81}$ | $33.11_{\pm3.17}$ | $33.78_{\pm1.76}$ | $39.02_{\pm1.30}$ | $49.17_{\pm4.41}$ | $47.93_{\pm4.43}$ | $53.13_{\pm2.29}$ |
| | **GACL** (ours) | ✓ | $\mathbf{57.99_{\pm2.46}}$ | $\mathbf{56.24_{\pm3.12}}$ | $\underline{\mathbf{70.31_{\pm0.06}}}$ | $\mathbf{41.68_{\pm0.78}}$ | $\mathbf{47.30_{\pm0.84}}$ | $\mathbf{42.22_{\pm0.10}}$ | $\mathbf{63.14_{\pm0.66}}$ | $\underline{\mathbf{69.32_{\pm0.87}}}$ | $\underline{\mathbf{62.68_{\pm0.08}}}$ |

# 4 Experiments

## 4.1 Experimental Setup

In the section, we conduct experiments on various benchmark datasets and compare the GACL with both EFCIL and replay-based state-of-the-art methods, including LwF [14], L2P [36], DualPrompt [33], ER [35], EWC++ [16], SLDA [37], RM [32], MVP [30], and MVP-R (MVP with exemplars).[3]

**Datasets.** We conduct experiments on three datasets: CIFAR-100 [38], ImageNet-R [39], and Tiny-ImageNet [40]. We evaluate each method under the Si-Blurry setting [30] (the most complex GCIL setting) with 5 independent seeds. For the Si-Blurry setting, we set the disjoint class ratio $r_{\text{D}}$ to $50\%$ and the blurry sample ratio $r_{\text{B}}$ to $10\%$. More details about Si-Blurry are listed in Appendix C.

**Implementation Details.** We utilize the DeiT-S/16 [41] as our backbone. Following [42, 43], we pre-train the backbone on 611 ImageNet classes after excluding 389 classes that overlap with CIFAR and Tiny-ImageNet to prevent data leakage. To ensure a fair comparison, all methods utilize a frozen backbone. All methods under comparison are implemented as specified in [30]. The memory sizes of compared relay-based methods are set to 500 and 2000.

There are two hyperparameters in the GACL, the regularization term $\gamma$ and the size of the buffer layer. Here, we adopt $\gamma = 100$, which is determined by the grid search of $\{0, 10, 100, 500, 1000, 10000\}$ on CIFAR-100 (by a 90%-10% train-val split). As the regularization term $\gamma$ is not sensitive in a proper range [7], we adopt this value for all datasets for convenience. We relocate its analysis to Appendix E. The size for the buffer layer $W_{\text{B}}$ is set to 5000 for both the GACL and ACIL for convenience.

**Evaluation Protocol.** Three metrics are adopted to evaluate GCIL tasks. The real-time performance is evaluated by the *area under the curve of accuracy* $\mathcal{A}_{\text{AUC}}$ [31], i.e., $\mathcal{A}_{\text{AUC}} = \sum_{i=1}^{k} f(i \cdot \triangle n) \cdot \triangle n$, where $\triangle n$ is the number of samples observed between evaluation and $f(\cdot)$ is the curve in the accuracy-to-{number of training samples} plot, measuring anytime inference performance during training. A higher $\mathcal{A}_{\text{AUC}}$ corresponds to a method consistently maintaining high accuracy throughout the training. The overall performance is evaluated by the *average incremental accuracy* (or average accuracy) $\mathcal{A}_{\text{Avg}} = \frac{1}{K+1} \sum_{k=1}^{K} \mathcal{A}_k$, where the task-wise accuracy $\mathcal{A}_k$ indicates the average test accuracy in task

---

[3]The results for MVP and MVP-R are based on their official implementation, committed on October 26, 2024 (commit ID: ad8d1426a497545ba634521c00008c34ceece799).

$k$ by testing the network on $\mathcal{D}_{1:k}^{\text{test}}$. A higher $\mathcal{A}_{\text{Avg}}$ score is preferred when evaluating algorithms. The last evaluation metric is the *last-task accuracy* $\mathcal{A}_{\text{Last}}$ evaluating the network's last-task performance after completing all tasks.

## 4.2 Comparison with State-of-the-arts

As shown in Figure 2, we comprehensively compare the GACL with both EFCIL and replay-based methods.

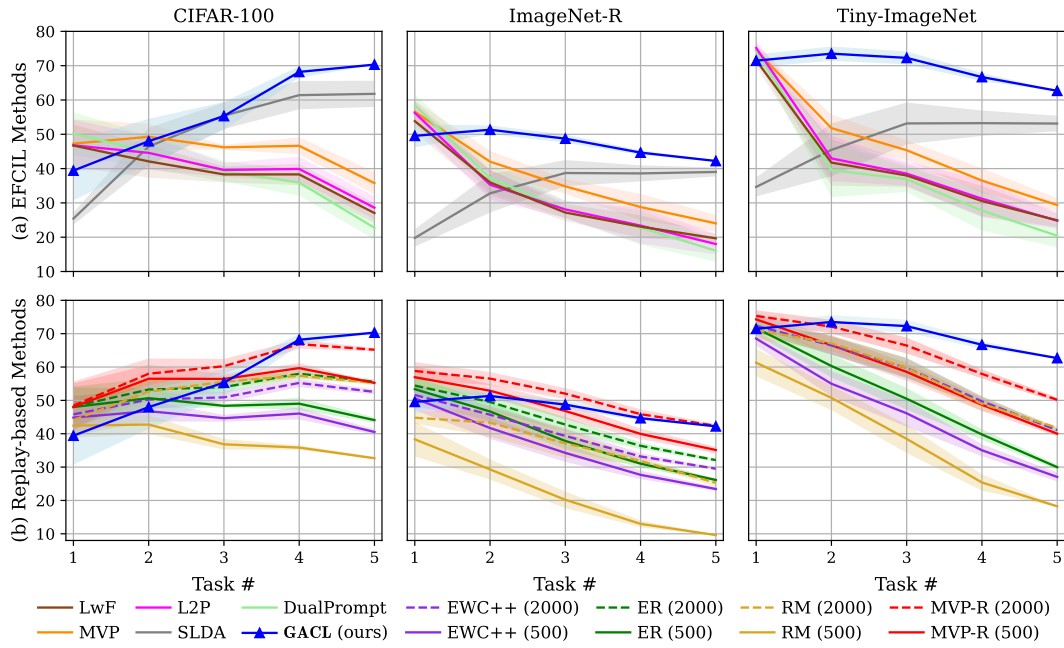

Figure 2: The task-wise accuracy $\mathcal{A}_k$ of the GACL with EFCIL methods (top) and replay-based methods (bottom) on benchmark datasets with the $K = 5$.

**Compare with EFCIL Methods.** EFCIL methods address privacy concerns and mitigate catastrophic forgetting without exemplars. Among EFCIL methods, our GACL consistently exhibits superior performance across all three datasets, as illustrated in the lower panel of Table 1.

For instance, on CIFAR-100, our method surpasses the second-best method SLDA, by **4.99**%, **6.15**%, and **8.52**% for $\mathcal{A}_{\text{AUC}}$, $\mathcal{A}_{\text{Avg}}$, and $\mathcal{A}_{\text{Last}}$, respectively. On Tiny-ImageNet, the GACL achieves impressive results with $\mathcal{A}_{\text{AUC}}$, $\mathcal{A}_{\text{Avg}}$, and $\mathcal{A}_{\text{Last}}$ reaching 63.14%, 69.32%, and 62.68%, respectively, surpassing the previous best EFCIL by **13.97**%, **21.39**%, and **9.55**%. Similar patterns are evident in the results of ImageNet-R, further confirming that the GACL is an exceptional tool for GCIL.

Owing to the weight-invariant property, the GACL exhibits more accurate and stable evolutions as $k$ increases as observed in Figure 2 (a). All compared EFCIL methods exhibit sharp declines in accuracy, while the GACL delivers nearly non-declining curves. In particular, on CIFAR-100, the GACL shows an unnatural improvement of task-wise accuracy throughout the learning tasks, with the GACL initially lagging behind other EFCIL methods. This is because the Si-Blurry samples more than 70% of the CIFAR-100 categories in the first two tasks (see Appendix F), constructing a scenario where gradient-based algorithms could largely avoid the forgetting issue. Moreover, our method produces more stable predictions across diverse scenarios, as indicated by much smaller standard errors (colored shades in Figure 2 (a)). In summary, the experimental results demonstrate that our proposed GACL is exceedingly accurate and robust, exhibiting exceptional generalization ability.

**Compare with Replay-based Methods.** Replay-based methods are considerably competitive as they leverage historical samples. The memory size is a key adjustment, as increasing it typically leads to performance improvements by allowing more historical knowledge to be reviewed. For instance, the MVP-R achieves 4.42%, 3.97%, and 8.95% gains for $\mathcal{A}_{\text{AUC}}$, $\mathcal{A}_{\text{Avg}}$, and $\mathcal{A}_{\text{Last}}$ (see Table 1) on CIFAR-100 when increasing the memory size from 500 to 2000.

As an exemplar-free technique, our GACL avoids re-using the historical samples. However, as indicated in Table 1, the GACL still outperforms most existing replay-based results. For instance, the GACL achieves the best $\mathcal{A}_{\text{Last}}$ results among all settings. The GACL's $\mathcal{A}_{\text{AUC}}$ and $\mathcal{A}_{\text{Avg}}$ results are also mostly superior, except that our performance is slightly weaker than that of MVP-R with a memory size of 2000 on CIFAR-100 and ImageNet-R. Although increasing the number of exemplars can further improve the results of replay-based methods, this approach could lead to higher training and memory costs and, more importantly, more severe privacy invasion.

As indicated in Figure 2 (b), replay-based methods experience accuracy declines similar to those observed in the EFCIL case. This decline is due to an inherent limitation of gradient-based iterative algorithms, which tend to favor recently trained categories and thus lead to catastrophic forgetting. The GACL is iterative-free and then not constrained by this forgetting issue, thereby achieving nearly no performance reduction as $K$ increases.

**Why the GACL Gives Leading Performance.** The above comparisons show that the proposed GACL is a powerful GCIL technique. Its competitive performance can be explained as follows. (i) Weight-invariant property. As shown in Theorem 3.1, the weight obtained recursively is equal to its joint-learning counterpart, indicating that the GACL is a "completely non-forgetting" technique (under the condition of a frozen backbone). (ii) Analytical solution. Existing GCIL techniques are gradient-based iterative algorithms prone to catastrophic forgetting by nature. The GACL is a new member of the ACL and inherits its non-iterative gradient-free essence with an analytical solution, thereby avoiding the task-recency bias to address forgetting.

## 4.3   Ablation Study on the ECLG Module

The ECLG module is a core component that allows the GACL to obtain the weight-invariant property in the GCIL scenario. Here, we conduct an ablation study to justify the ECLG's contributions under various blurry sample ratios $r_B$ with $r_{\text{D}} = 50\%$. Larger $r_B$ indicates more complex data distributions in the Si-Blurry setting. As shown in Table 2, the GACL without ECLG exhibits poor performance with a visible gap for $\mathcal{A}_{\text{AUC}}$, $\mathcal{A}_{\text{Avg}}$, and $\mathcal{A}_{\text{Last}}$. For instance, on CIFAR-100 with $r_B = 10\%$, the ECLG contributes a 23.01% $\mathcal{A}_{\text{Last}}$ gain to the GACL.

Table 2: Ablation study on the ECLG module of our GACL.

| $r_{\text{B}}$ | Dataset | With ECLG | | | Without ECLG | | |
|---|---|---|---|---|---|---|---|
| | | $\mathcal{A}_{\text{AUC}}(\%)$ | $\mathcal{A}_{\text{Avg}}(\%)$ | $\mathcal{A}_{\text{Last}}(\%)$ | $\mathcal{A}_{\text{AUC}}(\%)$ | $\mathcal{A}_{\text{Avg}}(\%)$ | $\mathcal{A}_{\text{Last}}(\%)$ |
| 10% | CIFAR-100 | $57.99_{\pm 2.46}$ | $56.24_{\pm 3.12}$ | $70.31_{\pm 0.06}$ | $45.68_{\pm 7.74}$ | $42.04_{\pm 4.52}$ | $47.30_{\pm 2.61}$ |
| | ImageNet-R | $41.68_{\pm 0.78}$ | $47.30_{\pm 0.84}$ | $42.22_{\pm 0.10}$ | $40.29_{\pm 2.23}$ | $46.95_{\pm 1.15}$ | $41.67_{\pm 0.36}$ |
| | Tiny-ImageNet | $63.14_{\pm 0.66}$ | $69.32_{\pm 0.87}$ | $62.68_{\pm 0.08}$ | $60.21_{\pm 1.86}$ | $65.80_{\pm 1.20}$ | $60.13_{\pm 0.37}$ |
| 30% | CIFAR-100 | $57.33_{\pm 1.03}$ | $58.74_{\pm 1.59}$ | $69.90_{\pm 0.01}$ | $42.53_{\pm 1.97}$ | $42.26_{\pm 1.75}$ | $45.49_{\pm 1.17}$ |
| | ImageNet-R | $42.19_{\pm 0.44}$ | $47.82_{\pm 1.11}$ | $42.90_{\pm 0.08}$ | $42.01_{\pm 0.26}$ | $46.95_{\pm 1.15}$ | $41.67_{\pm 0.56}$ |
| | Tiny-ImageNet | $60.73_{\pm 1.15}$ | $67.31_{\pm 1.14}$ | $59.73_{\pm 2.55}$ | $60.63_{\pm 1.86}$ | $57.03_{\pm 1.98}$ | $60.13_{\pm 0.55}$ |
| 50% | CIFAR-100 | $56.74_{\pm 1.14}$ | $58.29_{\pm 1.95}$ | $70.02_{\pm 0.05}$ | $40.91_{\pm 3.57}$ | $47.25_{\pm 2.64}$ | $58.61_{\pm 2.62}$ |
| | ImageNet-R | $41.33_{\pm 1.46}$ | $46.42_{\pm 2.30}$ | $42.92_{\pm 0.17}$ | $40.44_{\pm 3.14}$ | $42.50_{\pm 3.43}$ | $39.05_{\pm 1.65}$ |
| | Tiny-ImageNet | $60.96_{\pm 1.83}$ | $66.28_{\pm 2.69}$ | $62.24_{\pm 0.10}$ | $60.32_{\pm 4.20}$ | $60.70_{\pm 4.30}$ | $56.97_{\pm 1.89}$ |

As claimed in Theorem 3.1, the classifier without the ECLG module fails to absorb knowledge from joint classes in each task (i.e., classes that reappear), leading to substantial information loss under the GCIL setting. The GACL, equipped with the ECLG module, demonstrates competence in handling overlapping classes in realistic scenarios.

## 4.4   Robustness Analysis in Si-Blurry Setting

Here, we conduct a robust analysis by varying the disjoint class ratio $r_{\text{D}}$ and the blurry sample ratio $r_{\text{B}}$. The comparison happens among the GACL, the second-best EFCIL method SLDA, and the top-performing replay-based method MVP-R with a memory size of 500.

We evaluate our method under various $r_D$, including extreme cases where each task shares classes ($r_D = 0\%$) and traditional CIL scenarios ($r_D = 100\%$). Table 3 illustrates that our GACL consistently outperforms the compared methods (e.g., leads the SLDA by 2%-10%) and produces near-identical $A_{Last}$ values with varying $r_D$. This shows the accurate and robust traits of the GACL.

We also evaluate our method using various $r_B$ values, as shown in Table 4. Similar patterns observed here align with those in Table 3, further demonstrating the robustness of the proposed GACL, which delivers exceptional performance across different GCIL settings.

Table 3: The performance at different $r_D$ with $r_B = 10\%$ on CIFAR-100.

| $r_D$ | Method | $\mathcal{A}_{AUC}(\%)$ | $\mathcal{A}_{Avg}(\%)$ | $\mathcal{A}_{Last}(\%)$ |
|---|---|---|---|---|
| | SLDA [37] | $\mathbf{55.51}_{\pm\mathbf{1.93}}$ | $\mathbf{53.94}_{\pm\mathbf{0.92}}$ | $67.45_{\pm0.26}$ |
| 0% | MVP-R [30] | $53.49_{\pm1.40}$ | $50.73_{\pm0.37}$ | $60.54_{\pm2.03}$ |
| | **GACL (ours)** | $49.96_{\pm0.61}$ | $50.56_{\pm0.49}$ | $\mathbf{69.94}_{\pm\mathbf{0.09}}$ |
| | SLDA [37] | $53.00_{\pm3.85}$ | $50.09_{\pm2.77}$ | $61.79_{\pm3.81}$ |
| 50% | MVP-R [30] | $56.20_{\pm1.47}$ | $53.61_{\pm0.04}$ | $55.35_{\pm0.43}$ |
| | **GACL (ours)** | $\mathbf{57.99}_{\pm\mathbf{2.46}}$ | $\mathbf{56.24}_{\pm\mathbf{3.12}}$ | $\mathbf{70.31}_{\pm\mathbf{0.06}}$ |
| | SLDA [37] | $65.46_{\pm4.79}$ | $67.29_{\pm5.28}$ | $63.56_{\pm2.68}$ |
| 100% | MVP-R [30] | $68.43_{\pm0.28}$ | $68.04_{\pm1.48}$ | $53.14_{\pm0.72}$ |
| | **GACL (ours)** | $\mathbf{70.72}_{\pm\mathbf{0.32}}$ | $\mathbf{77.57}_{\pm\mathbf{1.02}}$ | $\mathbf{69.97}_{\pm\mathbf{0.03}}$ |

Table 4: The performance at different $r_B$ with $r_D = 50\%$ on CIFAR-100.

| $r_B$ | Method | $\mathcal{A}_{AUC}(\%)$ | $\mathcal{A}_{Avg}(\%)$ | $\mathcal{A}_{Last}(\%)$ |
|---|---|---|---|---|
| | SLDA [37] | $53.00_{\pm3.85}$ | $50.09_{\pm2.77}$ | $61.79_{\pm3.81}$ |
| 10% | MVP-R [30] | $56.20_{\pm1.47}$ | $53.61_{\pm0.04}$ | $55.35_{\pm0.43}$ |
| | **GACL (ours)** | $\mathbf{57.99}_{\pm\mathbf{2.46}}$ | $\mathbf{56.24}_{\pm\mathbf{3.12}}$ | $\mathbf{70.31}_{\pm\mathbf{0.06}}$ |
| | SLDA [37] | $54.55_{\pm4.66}$ | $54.06_{\pm2.41}$ | $63.04_{\pm2.56}$ |
| 30% | MVP-R [30] | $\underline{59.65}_{\pm2.04}$ | $58.31_{\pm1.52}$ | $58.16_{\pm1.38}$ |
| | **GACL (ours)** | $\mathbf{57.33}_{\pm\mathbf{1.03}}$ | $\underline{\mathbf{58.74}}_{\pm\mathbf{1.59}}$ | $\mathbf{69.90}_{\pm\mathbf{0.01}}$ |
| | SLDA [37] | $53.81_{\pm3.43}$ | $52.93_{\pm2.36}$ | $63.45_{\pm2.72}$ |
| 50% | MVP-R [30] | $\underline{59.10}_{\pm1.98}$ | $57.34_{\pm1.96}$ | $54.81_{\pm0.21}$ |
| | **GACL (ours)** | $\mathbf{56.74}_{\pm\mathbf{1.14}}$ | $\underline{\mathbf{58.29}}_{\pm\mathbf{1.95}}$ | $\mathbf{70.02}_{\pm\mathbf{0.05}}$ |

## 4.5 Limitation and Future Work

Overall, the GACL exhibits various good characteristics as an exemplar-free GCIL technique. The major limitation here is the need for a well-trained backbone because the GACL does not update backbone weights. This could motivate the exploration of adjustable backbones to continuously improve their feature extraction abilities, thereby further enhancing GACL's performance.

## 5 Conclusion

In this paper, we introduce the exemplar-free generalized analytic class incremental learning (GACL) approach to address the GCIL problem. Building upon analytic learning, the GACL delivers closed-form solutions to GCIL through the decomposition of GCIL data into exposed and unexposed classes. The GACL achieves the weight-invariant property that provides identical solutions for GCIL to its joint learning counterpart. We theoretically validate this property and provide high interpretability through the matrix analysis tool. Various experiments are conducted under the Si-Blurry setting, demonstrating that our proposed GACL achieves remarkable performance with high robustness compared to state-of-the-art EFCIL and replay-based methods.

## Acknowledgments and Disclosure of Funding

This research was supported by the National Natural Science Foundation of China (62306117), the Guangzhou Basic and Applied Basic Research Foundation (2024A04J3681, 2023A04J1687), the South China University of Technology-TCL Technology Innovation Fund, the Fundamental Research Funds for the Central Universities (2023ZYGXZR023, 2024ZYGXZR074), the Guangdong Basic and Applied Basic Research Foundation (2024A1515010220), the CAAI-MindSpore Open Fund developed on Openl Community, the Shenzhen Fundamental Research Program (JCYJ20230807091809020), and Shenzhen Science and Technology Plan (Grant No. JCYJ20210324123802006).

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

## A   Proof of Theorem 3.1

*Proof.* in task $k-1$, we have

$$\hat{\boldsymbol{W}}_{\text{FCN}}^{(k-1)} = (\boldsymbol{X}_{1:k-2}^{\text{total}\top}\boldsymbol{X}_{1:k-2}^{\text{total}} + \boldsymbol{X}_{k-1}^{(\text{B})\top}\boldsymbol{X}_{1:k-1}^{(\text{B})} + \gamma\boldsymbol{I})^{-1}\left[\boldsymbol{X}_{1:k-2}^{\text{total}\top}\boldsymbol{Y}_{1:k-2}^{\text{total}} + \boldsymbol{X}_{k-1}^{(\text{B})\top}\bar{\boldsymbol{Y}}_{k-1}^{\text{train}} \quad \boldsymbol{X}_{k-1}^{(\text{B})\top}\tilde{\boldsymbol{Y}}_{k}^{\text{train}}\right]. \tag{13}$$

Hence, in task $k$, we have

$$\hat{\boldsymbol{W}}_{\text{FCN}}^{(k)} = (\boldsymbol{X}_{1:k-1}^{\text{total}\top}\boldsymbol{X}_{1:k-1}^{\text{total}} + \boldsymbol{X}_{k}^{(\text{B})\top}\boldsymbol{X}_{k}^{(\text{B})} + \gamma\boldsymbol{I})^{-1}\left[\boldsymbol{X}_{1:k-1}^{\text{total}\top}\boldsymbol{Y}_{1:k-1}^{\text{total}} + \boldsymbol{X}_{k}^{(\text{B})\top}\bar{\boldsymbol{Y}}_{k}^{\text{train}} \quad \boldsymbol{X}_{k}^{(\text{B})\top}\tilde{\boldsymbol{Y}}_{k}^{\text{train}}\right]. \tag{14}$$

We have defined the autocorrelation memory matrix $\boldsymbol{R}_{k-1}$ in the paper via

$$\boldsymbol{R}_{k-1} = (\boldsymbol{X}_{1:k-2}^{\text{total}\top}\boldsymbol{X}_{1:k-2}^{\text{total}} + \boldsymbol{X}_{k-1}^{(\text{B})\top}\boldsymbol{X}_{k-1}^{(\text{B})} + \gamma\boldsymbol{I})^{-1}. \tag{15}$$

To facilitate subsequent calculations, here we also define a cross-correlation matrix $\boldsymbol{Q}_{k-1}$, i.e.,

$$\boldsymbol{Q}_{k-1} = \left[\boldsymbol{X}_{1:k-2}^{\text{total}\top}\boldsymbol{Y}_{1:k-2}^{\text{total}} + \boldsymbol{X}_{k-1}^{(\text{B})\top}\bar{\boldsymbol{Y}}_{k-1}^{\text{train}} \quad \boldsymbol{X}_{k-1}^{(\text{B})\top}\tilde{\boldsymbol{Y}}_{k}^{\text{train}}\right]. \tag{16}$$

Thus we can rewrite (13) as

$$\hat{\boldsymbol{W}}_{\text{FCN}}^{(k-1)} = \boldsymbol{R}_{k-1}\boldsymbol{Q}_{k-1}. \tag{17}$$

Therefore, in task $k$ we have

$$\hat{\boldsymbol{W}}_{\text{FCN}}^{(k)} = \boldsymbol{R}_{k}\boldsymbol{Q}_{k}. \tag{18}$$

From (15), we can recursively calculate $\boldsymbol{R}_{k}$ from $\boldsymbol{R}_{k-1}$, i.e.,

$$\boldsymbol{R}_{k} = \left(\boldsymbol{R}_{k-1}^{-1} + \boldsymbol{X}_{k}^{(\text{B})\top}\boldsymbol{X}_{k}^{(\text{B})}\right)^{-1}. \tag{19}$$

According to the Woodbury matrix identity, we have

$$(\boldsymbol{A} + \boldsymbol{U}\boldsymbol{C}\boldsymbol{V})^{-1} = \boldsymbol{A}^{-1} - \boldsymbol{A}^{-1}\boldsymbol{U}(\boldsymbol{C}^{-1} + \boldsymbol{V}\boldsymbol{A}^{-1}\boldsymbol{U})^{-1}\boldsymbol{V}\boldsymbol{A}^{-1}.$$

Let $\boldsymbol{A} = \boldsymbol{R}_{k-1}^{-1}$, $\boldsymbol{U} = \boldsymbol{X}_{k}^{(\text{B})\top}$, $\boldsymbol{C} = \boldsymbol{I}$, and $\boldsymbol{V} = \boldsymbol{X}_{k}^{(\text{B})}$ in (19), we have

$$\boldsymbol{R}_{k} = \boldsymbol{R}_{k-1} - \boldsymbol{R}_{k-1}\boldsymbol{X}_{k}^{(\text{B})\top}(\boldsymbol{I} + \boldsymbol{X}_{k}^{(\text{B})}\boldsymbol{R}_{k-1}\boldsymbol{X}_{k}^{(\text{B})\top})^{-1}\boldsymbol{X}_{k}^{(\text{B})}\boldsymbol{R}_{k-1}. \tag{20}$$

Hence, $\boldsymbol{R}_{k}$ can be recursively updated using its last-task counterpart $\boldsymbol{R}_{k-1}$ and data from the current task (i.e., $\boldsymbol{X}_{k}^{(\text{B})}$). This proves the recursive calculation of the autocorrelation memory matrix.

Next, we derive the recursive formulation of $\hat{\boldsymbol{W}}_{\text{FCN}}^{(k)}$. To this end, we also recurse the cross-correlation matrix $\boldsymbol{Q}_{k}$ in task $k$, i.e.,

$$\boldsymbol{Q}_{k} = \left[\boldsymbol{X}_{1:k-1}^{\text{total}\top}\boldsymbol{Y}_{1:k-1}^{\text{total}} + \boldsymbol{X}_{k}^{(\text{B})\top}\bar{\boldsymbol{Y}}_{k}^{\text{train}} \quad \boldsymbol{X}_{k}^{(\text{B})\top}\tilde{\boldsymbol{Y}}_{k}^{\text{train}}\right] = \boldsymbol{Q}_{k-1}' + \left[\boldsymbol{X}_{k}^{(\text{B})\top}\bar{\boldsymbol{Y}}_{k}^{\text{train}} \quad \boldsymbol{X}_{k}^{(\text{B})\top}\tilde{\boldsymbol{Y}}_{k}^{\text{train}}\right], \tag{21}$$

where

$$\boldsymbol{Q}_{k-1}' = \begin{cases} \left[\boldsymbol{Q}_{k-1} \quad \boldsymbol{0}_{d_{(\text{B})}\times(d_{y_k}-d_{y_{k-1}})}\right], & d_{y_k} > d_{y_{k-1}} \\ \boldsymbol{Q}_{k-1}, & d_{y_k} = d_{y_{k-1}} \end{cases}. \tag{22}$$

Note that the concatenation in (22) is due to the assumption that $\boldsymbol{Y}_{1:k}^{\text{train}}$ in task $k$ contains more data classes (hence more columns) than $\boldsymbol{Y}_{1:k-1}^{\text{train}}$. It is possible that there are no new classes appear in task $k$, then $\tilde{\boldsymbol{Y}}_{k}^{\text{train}}$ should be $\boldsymbol{0}$.

Similar to what (22) does,

$$\hat{\boldsymbol{W}}_{\text{FCN}}^{(k-1)\prime} = \begin{cases} \begin{bmatrix} \hat{\boldsymbol{W}}_{\text{FCN}}^{(k-1)} & \boldsymbol{0}_{d_{(\text{B})} \times (d_{y_k} - d_{y_{k-1}})} \end{bmatrix}, & d_{y_k} > d_{y_{k-1}} \\ \hat{\boldsymbol{W}}_{\text{FCN}}^{(k-1)}, & d_{y_k} = d_{y_{k-1}} \end{cases} \tag{23}$$

We have

$$\hat{\boldsymbol{W}}_{\text{FCN}}^{(k-1)\prime} = \boldsymbol{R}_{k-1} \boldsymbol{Q}_{k-1}'. \tag{24}$$

Hence, $\hat{\boldsymbol{W}}_{\text{FCN}}^{(k)}$ can be rewritten as

$$\begin{aligned} \hat{\boldsymbol{W}}_{\text{FCN}}^{(k)} &= \boldsymbol{R}_k \boldsymbol{Q}_k \\ &= \boldsymbol{R}_k (\boldsymbol{Q}_{k-1}' + \begin{bmatrix} \boldsymbol{X}_k^{(\text{B})\top} \bar{\boldsymbol{Y}}_k^{\text{train}} & \boldsymbol{X}_k^{(\text{B})\top} \tilde{\boldsymbol{Y}}_k^{\text{train}} \end{bmatrix}) \\ &= \boldsymbol{R}_k \boldsymbol{Q}_{k-1}' + \boldsymbol{R}_k \boldsymbol{X}_k^{(\text{B})\top} \begin{bmatrix} \bar{\boldsymbol{Y}}_k^{\text{train}} & \tilde{\boldsymbol{Y}}_k^{\text{train}} \end{bmatrix}. \end{aligned} \tag{25}$$

By substituting (20) into $\boldsymbol{R}_k \boldsymbol{Q}_{k-1}'$, we have

$$\begin{aligned} \boldsymbol{R}_k \boldsymbol{Q}_{k-1}' &= \boldsymbol{R}_{k-1} \boldsymbol{Q}_{k-1}' - \boldsymbol{R}_{k-1} \boldsymbol{X}_k^{(\text{B})\top} (\boldsymbol{I} + \boldsymbol{X}_k^{(\text{B})} \boldsymbol{R}_{k-1} \boldsymbol{X}_k^{(\text{B})\top})^{-1} \boldsymbol{X}_k^{(\text{B})} \boldsymbol{R}_{k-1} \boldsymbol{Q}_{k-1}' \\ &= \hat{\boldsymbol{W}}_{\text{FCN}}^{(k-1)\prime} - \boldsymbol{R}_{k-1} \boldsymbol{X}_k^{(\text{B})\top} (\boldsymbol{I} + \boldsymbol{X}_k^{(\text{B})} \boldsymbol{R}_{k-1} \boldsymbol{X}_k^{(\text{B})\top})^{-1} \boldsymbol{X}_k^{(\text{B})} \hat{\boldsymbol{W}}_{\text{FCN}}^{(k-1)\prime}. \end{aligned} \tag{26}$$

To simplify this equation, let $\boldsymbol{K}_k = (\boldsymbol{I} + \boldsymbol{X}_k^{(\text{B})} \boldsymbol{R}_{k-1} \boldsymbol{X}_k^{(\text{B})\top})^{-1}$. Since

$$\boldsymbol{I} = \boldsymbol{K}_k \boldsymbol{K}_k^{-1} = \boldsymbol{K}_k (\boldsymbol{I} + \boldsymbol{X}_k^{(\text{B})} \boldsymbol{R}_{k-1} \boldsymbol{X}_k^{(\text{B})\top}),$$

we have $\boldsymbol{K}_k = \boldsymbol{I} - \boldsymbol{K}_k \boldsymbol{X}_k^{(\text{B})} \boldsymbol{R}_{k-1} \boldsymbol{X}_k^{(\text{B})\top}$. Therefore,

$$\begin{aligned} &\boldsymbol{R}_{k-1} \boldsymbol{X}_k^{(\text{B})\top} (\boldsymbol{I} + \boldsymbol{X}_k^{(\text{B})} \boldsymbol{R}_{k-1} \boldsymbol{X}_k^{(\text{B})\top})^{-1} \\ &= \boldsymbol{R}_{k-1} \boldsymbol{X}_k^{(\text{B})\top} \boldsymbol{K}_k \\ &= \boldsymbol{R}_{k-1} \boldsymbol{X}_k^{(\text{B})\top} (\boldsymbol{I} - \boldsymbol{K}_k \boldsymbol{X}_k^{(\text{B})} \boldsymbol{R}_{k-1} \boldsymbol{X}_k^{(\text{B})\top}) \\ &= (\boldsymbol{R}_{k-1} - \boldsymbol{R}_{k-1} \boldsymbol{X}_k^{(\text{B})\top} \boldsymbol{K}_k \boldsymbol{X}_k^{(\text{B})} \boldsymbol{R}_{k-1}) \boldsymbol{X}_k^{(\text{B})\top} \\ &= \boldsymbol{R}_k \boldsymbol{X}_k^{(\text{B})\top}. \end{aligned} \tag{27}$$

Substituting (27) into (26), $\boldsymbol{R}_k \boldsymbol{Q}_{k-1}'$ can be written as

$$\boldsymbol{R}_k \boldsymbol{Q}_{k-1}' = \hat{\boldsymbol{W}}_{\text{FCN}}^{(k-1)\prime} - \boldsymbol{R}_k \boldsymbol{X}_k^{(\text{B})\top} \boldsymbol{X}_k^{(\text{B})} \hat{\boldsymbol{W}}_{\text{FCN}}^{(k-1)\prime}. \tag{28}$$

Substituting (28) into (25) implies that

$$\begin{aligned} \hat{\boldsymbol{W}}_{\text{FCN}}^{(k)} &= \hat{\boldsymbol{W}}_{\text{FCN}}^{(k-1)\prime} - \boldsymbol{R}_k \boldsymbol{X}_k^{(\text{B})\top} \boldsymbol{X}_k^{(\text{B})} \hat{\boldsymbol{W}}_{\text{FCN}}^{(k-1)\prime} + \boldsymbol{R}_k \boldsymbol{X}_k^{(\text{B})\top} \begin{bmatrix} \bar{\boldsymbol{Y}}_k^{\text{train}} & \tilde{\boldsymbol{Y}}_k^{\text{train}} \end{bmatrix} \\ &= \begin{bmatrix} \hat{\boldsymbol{W}}_{\text{FCN}}^{(k-1)} - \boldsymbol{R}_k \boldsymbol{X}_k^{(\text{B})\top} \boldsymbol{X}_k^{(\text{B})} \hat{\boldsymbol{W}}_{\text{FCN}}^{(k-1)} + \boldsymbol{R}_k \boldsymbol{X}_k^{(\text{B})\top} \bar{\boldsymbol{Y}}_k^{\text{train}} & \boldsymbol{R}_k \boldsymbol{X}_k^{(\text{B})\top} \tilde{\boldsymbol{Y}}_k^{\text{train}} \end{bmatrix}. \end{aligned} \tag{29}$$

which completes the proof.

$\square$

## B  GCIL Properties

The GCIL scenario [6] is a recent CIL focus. Given task-wise learning tasks, we can involve all class labels in a set $\mathcal{S}$ with the number of classes $N$. The sample sizes, such as the numbers of input images of different classes appearing in task $k$, are modeled as a random vector $\boldsymbol{c}_k \in \mathbb{R}^N$. Each entry $\boldsymbol{c}_{k,i}$ is a random variable denoting the sample size of class $i$ in task $k$. In the generalized form, $\boldsymbol{c}_k$ is sampled from a task-dependent distribution. The GCIL scenario can be summarized as the following three key properties.

**Property B.1.** *The number of classes in a task is not fixed. Suppose $m_k$ is the number of classes in task $k$, we have:*

$$M_k = |\{i \in \mathcal{S} : \boldsymbol{c}_{k,i} > 0\}| \sim \mathcal{M}_k, \tag{30}$$

*where $\mathcal{M}_k$ is a task-dependent distribution.*

**Property B.2.** *Classes appearing in different tasks could overlap. For two tasks $k$ and $k'$, $k \neq k'$, we have:*

$$P(\boldsymbol{c}_k \odot \boldsymbol{c}_{k'} \neq 0) > 0, \tag{31}$$

*where $\odot$ denotes element-wise multiplication of two vectors and $P(\cdot)$ is the probability.*

**Property B.3.** *Sample sizes of different classes at the same task could be different. That is, for task $k$, we have*

$$i, j \in \mathcal{S}, i \neq j, P(\boldsymbol{c}_{k,i} \neq \boldsymbol{c}_{k,j} \mid \boldsymbol{c}_{k,i} \neq 0, \boldsymbol{c}_{k,j} \neq 0) > 0. \tag{32}$$

In short, the number of classes and samples could vary throughout the continual learning.

## C  Si-Blurry Setting

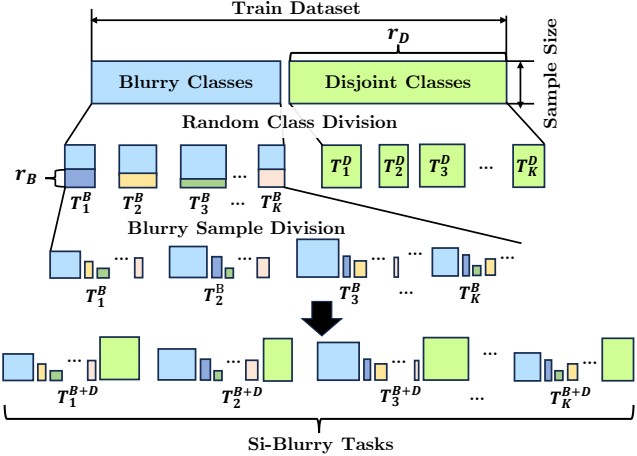

Figure 3: A configuration example of Si-Blurry setting.

The Si-Blurry setting [30] satisfies all the three properties of GCIL mentioned in Appendix B and can be treated as its good realization. As shown in Figure 3, for a $K$-task learning, the Si-Blurry first randomly partitions all classes into two groups: disjoint classes that cannot overlap between tasks and blurry classes that might reappear. The ratio of partition is controlled by the *disjoint class ratio* $\boldsymbol{r}_\mathrm{D}$, which is defined as the ratio of the number of disjoint classes to the number of all classes. Then disjoint classes and blurry classes are randomly assigned to disjoint tasks ($T^\mathrm{D}$) and blurry tasks ($T^\mathrm{B}$) respectively. Next, each blurry task further conducts the blurry sample division by randomly extracting part of samples to assign to other blurry tasks based on *blurry sample ratio* $\boldsymbol{r}_\mathrm{B}$, which is defined as the ratio of the extracted sample within samples in all blurry tasks. Finally, each Si-Blurry task $T^\mathrm{B+D}$ with a stochastic blurry task boundary consists of a disjoint and blurry task. We adopt Si-Blurry with different combinations of $\boldsymbol{r}_\mathrm{D}$ and $\boldsymbol{r}_\mathrm{B}$ for reliable empirical validations.

## D  Compute Resources

**GPU Usage**. We conduct experiments in PyTorch on one Nvidia Geforce RTX 4090 GPU with a batch size of 64 for training and 128 for inference. Figure 4 shows that the GACL uses minimal GPU memory. Our GACL significantly reduces GPU memory usage since it requires no back-propagation, thereby detaching gradients from tensors during calculations. This characteristic allows our approach to be applied with a larger batch size without memory leaks.

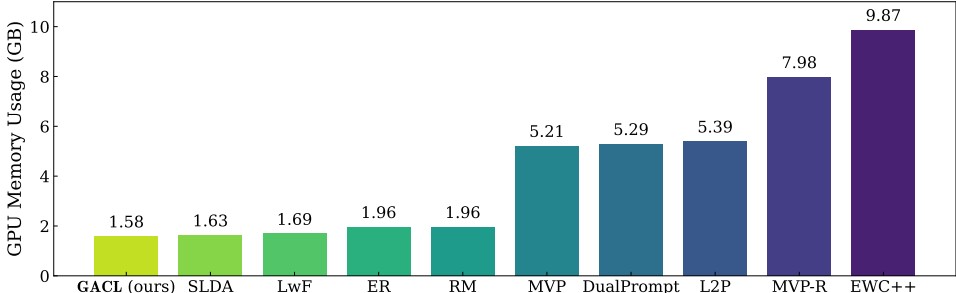

Figure 4: GPU memory consumption in GB with a batch size of 64 where replay-based methods are with 2000 memory size.

**Training Time**. Table 5 further illustrates the GACL's training time compared to others on one Nvidia Geforce RTX 4090 GPU, highlighting its efficiency. The GACL is faster than any other baselines except SLDA on three datasets because only the classifier and autocorrelation memory matrix $R$ are updated, leading to small numbers of trainable parameters compared to those baselines in a back-propagation manner.

Table 5: Average Training time of 5 independent seeds in seconds (s) where replay-based methods are with 2000 memory size.

| Method | EFCIL | CIFAR-100 (s) | ImageNet-R (s) | Tiny-ImageNet (s) |
|---|---|---|---|---|
| RM [32] | ✗ | >2 days | >2 days | >2 days |
| MVP-R [30] | ✗ | 717 | 527 | 1597 |
| ER [35] | ✗ | 369 | 330 | 715 |
| EWC++ [16] | ✗ | 650 | 391 | 1356 |
| LwF [14] | ✓ | 334 | 229 | 862 |
| L2P [36] | ✓ | 651 | 285 | 1246 |
| DualPrompt [33] | ✓ | 656 | 332 | 1294 |
| MVP [30] | ✓ | 628 | 300 | 1345 |
| SLDA [37] | ✓ | 401 | 284 | 915 |
| **GACL** (ours) | ✓ | **611** | **321** | **1246** |

## E  Hyperparameter Analysis for Regularization Term

Table 6: $\mathcal{A}_{\text{AUC}}$, $\mathcal{A}_{\text{Avg}}$, and $\mathcal{A}_{\text{Last}}$ of the GACL on all benchmark datasets with various values of the regularization term $\gamma$.

| $\gamma$ | CIFAR-100 (%) | | | ImageNet-R (%) | | | Tiny-ImageNet (%) | | |
|---|---|---|---|---|---|---|---|---|---|
| | $\mathcal{A}_{\text{AUC}}$ | $\mathcal{A}_{\text{Avg}}$ | $\mathcal{A}_{\text{Last}}$ | $\mathcal{A}_{\text{AUC}}$ | $\mathcal{A}_{\text{Avg}}$ | $\mathcal{A}_{\text{Last}}$ | $\mathcal{A}_{\text{AUC}}$ | $\mathcal{A}_{\text{Avg}}$ | $\mathcal{A}_{\text{Last}}$ |
| 0 | $8.87_{\pm 4.96}$ | $9.83_{\pm 5.82}$ | $8.65_{\pm 6.47}$ | $2.03_{\pm 0.36}$ | $2.85_{\pm 0.86}$ | $0.71_{\pm 0.09}$ | $4.38_{\pm 2.17}$ | $6.14_{\pm 4.01}$ | $0.62_{\pm 0.11}$ |
| 10 | $57.57_{\pm 2.35}$ | $55.97_{\pm 3.22}$ | $\mathbf{70.45_{\pm 0.08}}$ | $38.65_{\pm 0.69}$ | $44.38_{\pm 0.83}$ | $41.96_{\pm 0.10}$ | $62.74_{\pm 0.64}$ | $69.24_{\pm 0.79}$ | $62.73_{\pm 0.09}$ |
| 100 | $\mathbf{57.99_{\pm 2.46}}$ | $\mathbf{56.24_{\pm 3.12}}$ | $70.31_{\pm 0.06}$ | $41.68_{\pm 0.78}$ | $47.30_{\pm 0.84}$ | $42.22_{\pm 0.10}$ | $\mathbf{63.14_{\pm 0.66}}$ | $\mathbf{69.32_{\pm 0.87}}$ | $\mathbf{62.68_{\pm 0.08}}$ |
| 500 | $56.98_{\pm 2.61}$ | $55.46_{\pm 3.23}$ | $70.00_{\pm 0.02}$ | $\mathbf{42.92_{\pm 0.79}}$ | $\mathbf{49.01_{\pm 0.85}}$ | $\mathbf{42.70_{\pm 0.14}}$ | $62.90_{\pm 0.67}$ | $68.95_{\pm 0.88}$ | $62.41_{\pm 0.09}$ |
| 1000 | $56.03_{\pm 2.70}$ | $54.76_{\pm 3.31}$ | $69.61_{\pm 0.08}$ | $42.69_{\pm 0.80}$ | $48.90_{\pm 0.90}$ | $42.67_{\pm 0.16}$ | $61.96_{\pm 0.67}$ | $68.48_{\pm 0.83}$ | $62.10_{\pm 0.07}$ |
| 10000 | $51.01_{\pm 3.04}$ | $50.92_{\pm 3.62}$ | $66.38_{\pm 0.07}$ | $38.55_{\pm 0.85}$ | $45.16_{\pm 0.84}$ | $40.10_{\pm 0.19}$ | $57.54_{\pm 0.74}$ | $65.21_{\pm 0.70}$ | $59.55_{\pm 0.07}$ |

The regularization term $\gamma$ plays a crucial role and demonstrates robust behavior throughout our experiments. We assess the impact of the regularization term $\gamma$ in Table 6 and visualize the real-time accuracy of the GACL as it learns from training samples in Figure 5. Table 6 reveals the GACL's consistent performance across a broad range of $\gamma$ values, spanning from 10 to 10000. This highlights the versatility and robustness of our proposed GACL. However, as indicated in Figure 5, $\gamma$ of 10000 leads to slightly poorer performance because the ACL is prone to underfitting due to simple linear regression [44].

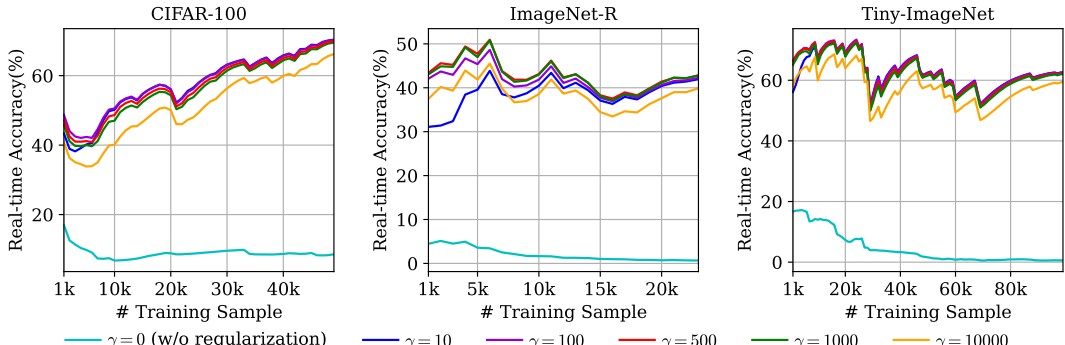

Figure 5: Real-time accuracy of the GACL on all benchmark datasets with various values of the regularization term $\gamma$.

Notably, both Table 6 and Figure 5 demonstrate that the absence of regularization results in a significant decline in performance. This underscores the crucial importance of incorporating $\gamma$ in the model. As indicated in (7), if we eliminate regularization by setting $\gamma$ to 0, the initial autocorrelation memory matrix $\boldsymbol{R}_0$ becomes zero. Subsequently, the computation of the autocorrelation memory matrix in task 1, denoted as $\boldsymbol{R}_1$, is expressed as:

$$\boldsymbol{R}_1 = (\boldsymbol{X}_1^{\text{total}\top}\boldsymbol{X}_1^{\text{total}})^{-1} = (\boldsymbol{X}_1^{(\text{B})\top}\boldsymbol{X}_1^{(\text{B})})^{-1}.$$

However, it's crucial to emphasize that $\boldsymbol{X}_1^{(\text{B})\top}\boldsymbol{X}_1^{(\text{B})}$ might result in a singular matrix, rendering it non-invertible. This potential singularity introduces an error in calculating $\boldsymbol{R}_1$, leading to a decrease in accuracy.

## F    Analysis of task-wise Accuracy Trends of the GACL

As depicted in Figure 2 (a), the task-wise accuracy of the GACL on CIFAR-100 demonstrates an increase. Notably, in the initial two tasks, the accuracy is lower compared to other EFCIL methods. However, on the other datasets, the GACL remains relatively stable. Upon a more detailed examination of the dataset split, we infer that the observed variations in trends are attributed to the specific dataset settings.

For a dataset with $N$ classes, the class number ratio $r_c$ after training on $i$-th samples is defined as $r_c = d_i/N$, where $d_i$ is the number of classes that have been seen observed at that point. As Figure 6 indicates, by examining the real-time accuracy and the class number ratio $r_c$ across the three sets of figures, a notable observation is made: when the sample size is small, the class number ratio $r_c$ on CIFAR-100 always surpasses that of the other two datasets on 5 seeds. This suggests that tasks on CIFAR-100 are notably more complex and intricate, resembling a few-shot learning scenario.

Consequently, the GACL exhibits lower task-wise accuracy compared to other gradient-based EFCIL methods, particularly in the initial stages. However, as more training samples are acquired, its accuracy progressively improves.

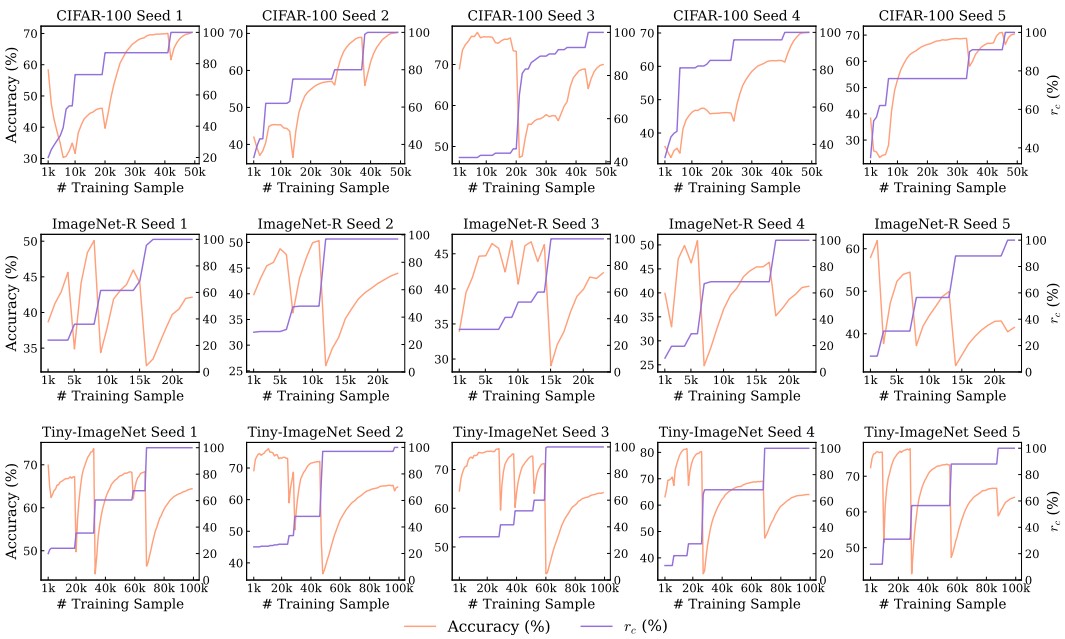

Figure 6: Real-time accuracy and class number ratio $r_c$ on 5 independent random seeds.

