# OpenReview forum: "GACL: Exemplar-Free Generalized Analytic Continual Learning"
_NeurIPS.cc/2024/Conference — NeurIPS 2024 poster_

### Official Review · Reviewer_tsV2 · 2024-07-03

**Soundness:** 4
**Presentation:** 3
**Contribution:** 4
**Rating:** 8
**Confidence:** 5

**Summary:**

This paper proposes a new exemplar-free GCIL technique named generalized analytic continual learning (GACL). It adopts analytic learning (a gradient-free training technique), and delivers an analytical (i.e., closed-form) solution to the GCIL scenario, which is derived via decomposing the incoming data into exposed and unexposed classes. The GACL attains a weight-invariant property, supporting an equivalence between the incremental learning and its joint training. This method is both therectically and empircally validated.

**Strengths:**

1. This is very interesting technique that obtains an equivalence between the GCIL and the joint training. Although a pre-trained network is needed, the weight-invariant property is very valuable in this area.
2. Another key contribution of GACL is that it is both accurate and exemplar-free, and this is powerful and significant.
3. The approach is also well-motivated and clearly explained.
4. The paper provides a comprehensive literature review, effectively contextualizing its contribution and facilitating the reader's understanding.
5. It is very clear theoretically regarding how the algorithm is parted into W_unexposed and W_ECLG.

**Weaknesses:**

1. Could you explain why the avg accuracy on CIFAR-100 is much lower than the last accuracy?
2. You need to explain more regarding "weight-invariance" in Theorem 3.1. It is not very easy to follow for those who are not in this area of research.
3. In this paper, the authors assume that the pre-trained backbone is generalizable so that the model can be frozen throughout the learning process. However, there are also some cases where the pre-trained model cannot generalize to downstream tasks. It would also be interesting to see the scenario where the pre-trained backbone yields a large domain gap.

**Questions:**

1. Continual learning has many variations (e.g., domain/task/class -incremental learning). Since the experiments only focus on generalized class-incremental learning, it would be better to adjust the title accordingly.
2. Since generalized CIL is also a specific case of class-incremental learning, it would be better to discuss why the proposed method is specially designed for GCIL. For example, when the data stream only contains new classes, will GCIL still work?
3. As the authors discussed in the main paper, the proposed method is a super case of ACIL and other analytical methods. As many readers would like to know, the authors are suggested to compare this method to other analytical methods to show how this method stands against other analytical works.

**Limitations:**

Yes

---

> ### Author Rebuttal · Authors · 2024-08-06
>
> # Replies to Reviewer tsV2
> Thank you for your constructive and detailed feedback. We provide detailed responses to your concerns
> below.
>
>
> ### W1. Why the avg accuracy on CIFAR-100 is much lower than the last accuracy?
> **Response to W_1**: As mentioned in Appendix G, tasks on CIFAR-100 are notably more complex and intricate, resembling a few-shot learning scenario. Consequently, the accuracy in the initial tasks tends to be lower, leading to a relatively low average accuracy.
>
> ### W2. Explain more regarding "weight-invariance" in Theorem 3.1.
> **Response to W_2**: The weights of the classifier obtained by this recursive update are exactly the same as the weights obtained by training the classifier from scratch on the entire data set. This *weight-invariant property* achieves a near “complete non-forgetting” and makes the GACL outperforms all EFCIL methods and most replay-based methods.
>
> ### W3. It would also be interesting to see the scenario where the pre-trained backbone yields a large domain gap.
> **Response to W_3**: As a demonstration, a 5-phase experiment is conducted in table below, with DeiT-S/16 backbone pretrained on ImageNet-1k, and CIL with DTD [1].
> |                        | Buffer | DTD |         |          |
> |:----------------------:|:------:|:--------:|:-------:|:--------:|
> |                        |        | ACC_AUC  | ACC_AVG | ACC_LAST |
> | EWC++ (ECCV,2018)     |  2000  |        55.66       |      58.97       |    47.32          |
> | ER (ICML,2019)        |  2000  |       51.78        |      57.87         |    50.99           |
> | MVP (ICCV,2023)       |  2000  |       52.97       |      55.15        |    54.10         |
> | EWC++ (ECCV,2018)     |  500   |      54.28        |     57.82          |    45.44          |
> | ER (ICML,2019)        |  500   |      51.23        |     57.45         |     49.67          |
> | MVP (ICCV,2023)       |  500   |      52.76        |     55.19       |       53.72        |
> | LwF (ECCV,2016)       |   0    |      42.67         |    48.41          |     33.42          |
> | L2P (CVPR 2022)       |   0    |     43.11          |    51.25           |    39.61           |
> | DualPrompt (ECCV 2022)|   0    |     42.12        |     50.17        |        39.29       |
> | MVP (ICCV,2023)       |   0    |     46.88       |      49.98       |         44.41       |
> | SLDA (IEEE/CVF2020)   |   0    |       49.84       |        50.33      |      55.11         |
> | **GACL**                   | **0**      | **61.93**    | **66.16**   | **63.69**    |
>
> [1] Mircea Cimpoi, Subhransu Maji, Iasonas Kokkinos, Sammy Mohamed, Andrea Vedaldi; Proceedings of the IEEE Conference on Computer Vision and Pattern Recognition (CVPR), 2014, pp. 3606-3613.
>
> ### Q1. Since the experiments only focus on generalized class-incremental learning, it would be better to adjust the title accordingly.
> **Response to Q_1**: Thank you for pointing out that. We will modify our title  into "Exemplar-Free Generalized Analytic Class Incremental Learning".
>
> ### Q2. Why is the proposed method specifically designed for generalized class-incremental learning (GCIL)?
> **Response to Q_2**: As demonstrate in Section 3.2 and 3.3, we introduce the ECLG module that corresponds to the exposed-class gain since classes can reappear in GCIL setting.
>
> ### Q3:Will GCIL still work if the data stream contains only new classes?
> **Response to Q_2**: Yes. We employ Si-Blurry[2] to generate GCIL tasks for our experiments. As demonstrated in Section 4.4, when the disjoint class ratio $r_{\text{D}}$ is 0, it indicates that the data stream contatians only new classes.
>
> [2] Jun-Yeong Moon, Keon-Hee Park, Jung Uk Kim, Gyeong-Moon Park; Proceedings of the IEEE/CVF International Conference on Computer Vision (ICCV), 2023, pp. 11731-11741

---

> ### Comment · Reviewer_tsV2 · 2024-08-11
>
> I thank the authors for providing the rebuttal. After reading the rebuttal and other reviewers' comments, all my previous concerns have been adequately addressed. I will keep my positive rating.

---

> > ### Author Response · Authors · 2024-08-12
> > **Thank you for the response**
> >
> > Thank you for taking the time to read our response! We are glad that our response addressed your concerns!

---

### Official Review · Reviewer_qU32 · 2024-07-09

**Soundness:** 3
**Presentation:** 3
**Contribution:** 2
**Rating:** 7
**Confidence:** 4

**Summary:**

Class Incremental Learning (CIL) faces the problem of catastrophic forgetting when training a network, i.e., the model loses previous knowledge when learning a new task. Generalized CIL (GCIL) aims to address more realistic scenarios, but existing methods are either ineffective or violate data privacy. This paper propose a new exemplar-free GCIL technique called Generalized Analytic Continual Learning (GACL), which provides a closed-form solution through analytic learning. GACL achieves equivalence between incremental learning and co-training by disaggregating the input data to keep the weights constant. This approach is theoretically validated and performs well across multiple datasets and settings.

**Strengths:**

* The paper is well-written and easy to follow.
* The proposed method outperforms other methods on several GCIL benchmarks.
* This paper provides a detailed proof of the theorem.

**Weaknesses:**

* The experimental setup is not clear. Given that the most recent method MVP [1] compared in this paper is the GCIL method for online CIL, were the experiments in this paper also conducted in an online scenario?
* The technical improvements on the previous work of ACL are somewhat incremental.

[1] Jun-Yeong Moon, Keon-Hee Park, Jung Uk Kim, and Gyeong-Moon Park. Online class incremental learning on stochastic blurry task boundary via mask and visual prompt tuning. ICCV 2023.

**Questions:**

1. What was the motivation for introducing ACL into GCIL? The claims in this paper seem to be an extended experiment of ACL in the GCIL scenario. It is difficult to generate further inspiration to the reader for solving the GCIL problem.
2. The proposed method utilize the DeiT-S/16 as backbone but previous works typically use ViT-B/16. Regarding the comparison methodology, what backbone network was used to obtain the reported results?
3. What are the technical improvements and contributions of this paper over previous ACL approaches such as ACIL [2] and DS-AL [3]?

[2] Huiping Zhuang, Zhenyu Weng, Hongxin Wei, RENCHUNZI XIE, Kar-Ann Toh, and Zhiping Lin. ACIL: Analytic class-incremental learning with absolute memorization and privacy protection. NeurIPS 2022
[3] Huiping Zhuang, Run He, Kai Tong, Ziqian Zeng, Cen Chen, and Zhiping Lin. DS-AL: A dual-stream analytic learning for exemplar-free class-incremental learning. AAAI 2024.

**Limitations:**

It is hard to foresee any potential negative societal impact of this theoretical work.

---

> ### Author Rebuttal · Authors · 2024-08-06
>
> # Replies to Reviewer qU32
> Thank you for your constructive and detailed feedback. We provide detailed responses to your concerns
> below.
>
> ### W1. Were the experiments in this paper also conducted in an online scenario?
> **Response to W_1**: Yes, we follow the settings in Si-Blurry [1], which is a online setting with blurry task boundaries.
>
> [1] Jun-Yeong Moon, Keon-Hee Park, Jung Uk Kim, Gyeong-Moon Park; Proceedings of the IEEE/CVF International Conference on Computer Vision (ICCV), 2023, pp. 11731-11741
>
> ### W2. "The technical improvements on the previous work of ACL are somewhat incremental." and "What are the technical improvements and contributions of this paper over previous ACL approaches such as ACIL and DS-AL?"
> **Response to W_2 and  Q_3**: We are very sorry that our delivery of GACL gives a "trivial extension" impression. In fact, the GACL is developed with nontrivial and sophisticated manupulations beyond existing ACL techniques. According to the derivation in lines 457-484, the solution **CAN NOT** be trivially extended from ACIL or its variants (the derivation takes **2 whole pages** in the appenix). We **deliberately** show the connection to the existing ACL techniques, to **better illustrate** the weight-invariant property (which has been recognied in ACIL, GKEAL). We will mark additional highlight in this part to avoid such an "incremental" impression.
>
> ### Q1. the motivation for introducing ACL into GCIL.
> **Response to Q_1**: Many real-world datasets follow GCIL tasks, such as the autonomous driving dataset SODA10M [2], the IoT dataset described in [3], and the real-world e-commerce service dataset described by [4]. GCIL simulates real-world incremental learning, where the distributions of data categories and sizes can be unknown in a given task. ACL techniques emerging as a new CIL branch, have a very appealing weight-invariant property in the CIL community, which **works magically esepcially for large-phase CIL tasks**. GCIL tasks (e.g., Si-blurry setting) are usually online (very large-phase), where ACL techinique can thrive. Existing ACL methods are specifically designed for CIL tasks only, and **CAN NOT** process GCIL ones, and the extension is **NOT** trivial (see derivations in lines 457-484), hence the motivation of GACL.
>
> [2] Han, Jianhua, et al. "SODA10M: A large-scale 2D self/semi-supervised object detection dataset for autonomous driving." *arXiv preprint arXiv:2106.11118* (2021).
>
> [3] Wen, Zhenyu, et al. "Fog orchestration for IoT services: issues, challenges and directions." *IEEE Internet Computing* 21.2 (2017): 16-24.
>
> [4] Bang, Jihwan, et al. "Rainbow memory: Continual learning with a memory of diverse samples." *Proceedings of the IEEE/CVF conference on computer vision and pattern recognition*. 2021.
>
> ### Q2. What backbone network was used to obtain the reported results?"
> **Response to Q_2**: As mentioned in Section 4.1, all experiments were conducted using the same pre-trained backbone. We utilize the DeiT-S/16 as our backbone and pre-train the backbone on 611 ImageNet classes after excluding 389 classes that overlap with CIFAR and Tiny-ImageNet to prevent data leakage.

---

> ### Comment · Reviewer_qU32 · 2024-08-09
> **Official Comment by Reviewer**
>
> Thanks to the authors for the response. This rebuttal addresses my concerns well. It fully explains the motivation for introducing ACL to address GCIL and the improvements this paper makes to existing ACL methods in response to the GCIL problem. Taking into account the comments of other reviewers and the authors' rebuttal, I decide to increase my rating.

---

> > ### Author Response · Authors · 2024-08-09
> > **Thank you for the response**
> >
> > Thank you for taking the time to read our response and increasing your score! We are glad to hear that the response addressed your concern.

---

### Official Review · Reviewer_Xb3R · 2024-07-11

**Soundness:** 3
**Presentation:** 4
**Contribution:** 2
**Rating:** 6
**Confidence:** 4

**Summary:**

This paper proposes a new exempler-free generalized continual learning (GCIL), named generalized analytic continual learning (GACL) technique. It does not depend on gradient-based tranining, which avoids the task-recency bias leading to the forgetting issue. It also delivers an closed-form solution to the GCIL scenario that provides identical solutions to its joint training. Entensive experiments demonstrate that the proposed GACL achieves consistently leading performance.

**Strengths:**

- The paper proposes very effective method that completely avoids forgetting in the gradient-free training technique.
- Extensive experiments are included explaining that GACL works good overall in the three datasets compared with many studies.
- Ablation studies are coducted that clarifies the differences among datasets and the reason why GACL is slightly worse than other baselines at early tasks. They also analyzed the contributions of ECLG module and its robustness.
- The paper is well written and easy to read.

**Weaknesses:**

- Missing baselines. As described in the Section 2, many ACL methods have been proposed. Comparing GACL with those methods (especially RanPAC) derives more solid results.
- Comparison to the joint-traning is not included. Although it is argued that GACL can bridge the gap between continual learning and joint-training, such result is not provided in the experiments.

**Questions:**

- I believe that the proposed gradient-free training scheme is beneficial also in the perspective of training speed. Does GACL run more fast than other (ACL) methods?

**Limitations:**

- As described, GACL cannot update backbone weights, which can be a problem. However, as new backbones emerge frequently, showing that GACL can work well not only with the used DeiT-S/16 but also others makes the arguments more convincing, instead.

---

> ### Author Rebuttal · Authors · 2024-08-06
>
> # Replies to Reviewer Xb3R
> Thank you for your constructive and detailed feedback. We provide detailed responses to your concerns
> below.
>
>
> ### 1. Comparing GACL with those methods (especially RanPAC)derives more solid results."
> **Response to W_1**: Thank you for pointing out this important reference as they both adopt closed-form solutions! The RanPAC emphasizes on random project while our GACL focuses on recursive formulation. As RanPAC's codes are for CIL tasks, and our settings (e.g., SI-Blurry) are very different, and they utilize different backbones, we are working on reproducing its results while facing several technical issues. We are sorry that we may miss the rebuttal deadline before providing the comparison. We will try to provide them during the disucssion period. Thanks!
>
> ### 2. Comparison to the joint-traning. Although it is argued that GACL can bridge the gap between continual learning and joint-training, such result is not provided in the experiments.
> **Response to W_2**:
>
> As suggested, we have included the last-phase accuracy of GACL and its joint-training accuracy on the CIFAR-100 dataset with different buffer sizes. Results in the table below show that the accuracy of GACL is indeed the same as that of joint training (the mile differences are caused by quantization).
>
> | Buffer Size | Joint-training (%) | G-ACL (%) |
> | :---------: | :----------------: | :-------: |
> |    1000     |      $66.83\pm0.14$      | $66.73\pm0.16$ |
> |    2000     |      $69.22\pm0.14$      | $69.22\pm0.23$ |
>
>
> ### 3. Does GACL run more fast than other (ACL) methods?
> **Response to Q_1**: In general, our GACL gives rather strong performance regarding speed. The table below further records the GACL's training time in seconds compared with EFCIL methods and replay-based methods with the memory size of 2000.
>
> The GACL runs faster than many baselines except a few candidates such as SLDA on three dataset. For SLDA, since only the classifier and autocorrelation memory matrix $\mathbf{R}$ are updated, leading to smaller numbers of trainable parameters compared with those baselines in back-propagation manner.
> | Methods      | EFCIL               | CIFAR-100 (s) | ImageNet-R (s) | Tiny-ImageNet (s)|
> |--------------|-------------|-----------|------------|---------------|
> | EM        | ❌     |  >2 days | >2 days        |    >2 days |
> | MVP-R        | ❌     |   717 | 527        |    1597 |
> | ER           | ❌       | 369       | 330        |     715   |
> | EWC++        | ❌       |  650  |     391   |      1356    |
> | LwF          | ✅        | 334       | 229        | 862           |
> | L2P          | ✅       |   651    |     285       |     1246     |
> | DualPrompt   | ✅       |    656    |    332  |     1294     |
> | MVP          | ✅         | 628       | 300        | 1345          |
> | SLDA         | ✅         | 401       | 284        | 915           |
> | GACL     | ✅       | 611   | 321    | 1246      |

---

> > ### Author Response · Authors · 2024-08-12
> > **Comparison update**
> >
> > Dear Reviewer,
> >
> > We would like to provde our updates during the discussion period, in which we have still been working on the response regarding the comparion with the RanPAC.
> >
> > We have managed to produce the results in comparison with RanPAC as follows.
> >
> > Overall, the RanPAC is a very strong-performing counterpart with average and final results comparable to ours. However, it is **NOT designed for GCIL/online tasks**, and hence **gives limited performance in ACC_AUC** (area under the curve of accuracy), which measures the online performance. For instance, on CIFAR-100, the GACL obtains 66.79% while RanPAC only has 51.62%.
> >
> > |                        | Buffer | CIFAR100 |         |         | ImageNet-R |         |         |  Tiny-ImageNet  |         |         |
> > |:----------------------:|:------:|:--------:|:-------:|:-------:|:-------------:|:-------:|:-------:|:-------------:|:--------:|:-------:|
> > |                        |        | ACC_AUC | ACC_AVG | ACC_LAST |ACC_AUC | ACC_AVG | ACC_LAST |ACC_AUC |ACC_AVG | ACC_LAST |
> > | RanPAC (NeurIPS,2023)     |  0  |  51.62 |     63.45 |  77.83  |  42.39  |  61.68  |   57.77  |  62.80  |  82.80  | 78.54  |
> > | GACL               | 0   | 66.79   | 63.94    | 77.34 |57.02      |62.26      |57.68      |77.65      |82.95      |77.80     |
> >
> > Let us know if more clarification is required!

---

### Official Review · Reviewer_vKCe · 2024-07-12

**Soundness:** 3
**Presentation:** 3
**Contribution:** 2
**Rating:** 5
**Confidence:** 4

**Summary:**

This paper deals with the generalized CIL (GCIL) problem where incoming data have mixed data categories and unknown sample size distribution. The author proposes generalized analytic continual learning (GACL) which adopts a pre-trained and fixed backbone and uses least squares to get a closed-form solution. Experiments verified that the GACL achieves better performance compared to other baselines.

**Strengths:**

* The generalized class incremental learning is an important and valuable problem.
* This paper is generally clear and easy to follow.
* Experiments show that GACL is effective in addressing forgetting in GCIL.

**Weaknesses:**

* Main concern. Most of the content on page 4 and page 5 (e.g., Theorem 3.1) in Section 3 is overly similar to existing ACL works [7, 8]. The main difference claimed by the authors, i.e., the ECLG module that corresponds to the exposed-class gain, is a trivial extension of the original ACL [7].
* Experiments. This paper adopts the pre-trained backbone developed in [40, 41]. However, many baselines (Table 2 in [40]) are missing in experiments. The GACL method is similar to RanPAC [25]. Experiments are needed to compare the performance of GACL with RanPAC.
* Comparison of the memory cost of weights in the buffer layer and other replay-based method.

**Questions:**

* How and why the dimensionality of the random projection affects the performance.

**Limitations:**

Yes.

---

> ### Author Rebuttal · Authors · 2024-08-06
>
> # Replies to Reviewer vKCe
> Thank you for your constructive and detailed feedback. We provide detailed responses to your concerns
> below.
>
> ### W1. Most of the content on page 4 and page 5 (e.g., Theorem 3.1) in Section 3 is overly similar to existing ACL works. The main difference claimed by the authors, i.e., the ECLG module that corresponds to the exposed-class gain, is a trivial extension of the original ACL."
>
> **Response to W_1**: Thank you for your valuable comment! We ackownledge that there are certain simlarities between ACIL and the GACL, mostly in the begnining part. However, this part is to **generate embedding** (e.g., Eq. 2, Eq. 3) and the **joint-learning objective function formulation** (e.g., Eq.4, Eq. 5), both of which are **NOT** the contributions of this paper. There are more like a **Preliminary** contents (e.g., like introducing what is loss function before designing a new one). Our key contributions start from Eq. 6 and Eq. 8 that separate the derivation into an exposed-unexposed pair. The derivation holds the key difference from ACL techniques (see lines 457-484). We apologize that appendixing these items have led to such confusion, and will try to highlight the difference by moving some of them in the main context.
>
> Regarding the concern of the ECLG module being a trivial extension, we respectfully disagree. According to the derivation in lines 457-484, the solution **CAN NOT** be trivially extended from ACIL or its variants. We **deliberately extracted the ECLG** to show the connection to the existing ACL techniques, to **better illustrate** the weight-invariant property (which has been recognied in ACIL, GKEAL). We are very sorry to receive the "trivial extension" impression, and will mark additional highlight in this part to avoid such an impression.
>
> ### W2. Experiments are needed to compare the performance of GACL with RanPAC.
>
> **Response to W_2**: Thank you for pointing out this important reference as they both adopt closed-form solutions! The RanPAC emphasizes on random project while our GACL focuses on recursive formulation. As RanPAC's codes are for CIL tasks, and our settings (e.g., SI-Blurry) are very different, and they utilize different backbones, we are working on reproducing its results while facing several technical issues. We are sorry that we may miss the rebuttal deadline before providing the comparison. We will try to provide them during the disucssion period!
>
>
> ### W3. Comparison of the memory cost of weights in the buffer layer and other replay-based method.
>
> **Response to W_3**: GACL operates without the need for prior knowledge regarding the total number of classes. We denote the size of features acquired after the feature extractor and buffer layer as "feature_size," and the number of classes learned at phase $k$ as "n_class." During the continual learning process, the classifier can progressively increase the number of classes in the head and simultaneously augment the dimension of $W_k$ (feature_size $\times$ n_class), while maintaining the size of $R_k$ as **feature_size $\times$ feature_size**.
>
> Compared with other replay-based methods, such as MVP [1] with a memory size of 500, the memory requirement is 500 $\times$ feature_size $\times$ feature_size $\times$ n_class, which **far exceeds the storage needed by GACL**. This is **without even accounting for model parameters and prompts**.
>
> [1] Jun-Yeong Moon, Keon-Hee Park, Jung Uk Kim, Gyeong-Moon Park; Proceedings of the IEEE/CVF International Conference on Computer Vision (ICCV), 2023, pp. 11731-11741
>
> ### W4. How and why the dimensionality of the random projection affects the performance.
>
> **Response to Q_1**: Based on Cover's theorem [2], a promising approach to enhancing the separability of features from different domains is to project the features extracted by the pre-trained model into a higher-dimensional space using a non-linear projection. This non-linear, higher-dimensional projection can improve performance by increasing the separability of the features.
>
> [2] T. M. Cover, "Geometrical and Statistical Properties of Systems of Linear Inequalities with Applications in Pattern Recognition," in IEEE Transactions on Electronic Computers, vol. EC-14, no. 3, pp. 326-334, June 1965, doi: 10.1109/PGEC.1965.264137.

---

> > ### Author Response · Authors · 2024-08-12
> > **More updates**
> >
> > Dear Reviewer,
> >
> > We would like to provde our updates during the discussion period, in which we have still been working on the response.
> >
> > 1) We have managed to produce the results in comparison with RanPAC as follows.
> >
> > Overall, the RanPAC is a very strong-performing counterpart with average and final results comparable to ours. However, it is **NOT designed for GCIL/online tasks**, and hence **gives limited performance in ACC_AUC** (area under the curve of accuracy), which measures the online performance. For instance, on CIFAR-100, the GACL obtains 66.79% while RanPAC only has 51.62%.
> >
> > |                        | Buffer | CIFAR100 |         |         | ImageNet-R |         |         |  Tiny-ImageNet  |         |         |
> > |:----------------------:|:------:|:--------:|:-------:|:-------:|:-------------:|:-------:|:-------:|:-------------:|:--------:|:-------:|
> > |                        |        | ACC_AUC | ACC_AVG | ACC_LAST |ACC_AUC | ACC_AVG | ACC_LAST |ACC_AUC |ACC_AVG | ACC_LAST |
> > | RanPAC (NeurIPS,2023)     |  0  |  51.62 |     63.45 |  77.83  |  42.39  |  61.68  |   57.77  |  62.80  |  82.80  | 78.54  |
> > | GACL               | 0   | 66.79   | 63.94    | 77.34 |57.02      |62.26      |57.68      |77.65      |82.95      |77.80     |
> >
> > 2) Regrarding your concern of novelty
> >
> > we have clarified this in "**Response to W_1**" in our rebuttal. Reviewer qU32 also pointed out this issue, and **has aknowledged our clarification**. Could you have a look and let us know if more is required to clarify this?

---

> > > ### Comment · Reviewer_vKCe · 2024-08-13
> > >
> > > I appreciated the author's responses and have raised my score to 5.

---

> > > > ### Author Response · Authors · 2024-08-14
> > > > **Thank you for the response**
> > > >
> > > > Thank you for taking the time to read our response! We are glad that our response addressed your concerns.

---

### Author Rebuttal · Authors · 2024-08-06

# General Response

We thank all the reviewers for their time, insightful suggestions and valuable comments. In summary, Reviewer vKCe, Reviewer Xb3R and Reviewer qU32 all appreciate that our writing is **clear** and **easy to follow**. Reviewer tsV2 appreciates that our method is **clear**, **well-motivated** and **valuable**.

We provide point-by-point responses to all reviewers’ comments and concerns. On the other hand, reviewers also point out that our delivery of GACL could resenble existing ACL techniques, giving an unnecessiry "trivial extension" impression. In this regard, we will marke the highlights of our GACL from formulation, derivation and contributions to avoid such an impression.

---

### Decision · Program_Chairs · 2024-09-25

**Decision:**

Accept (poster)

**Comment:**

The paper addresses the problem of a generalized form of lifelong learning, where learning is pursued in a continuous manner, with the possibility of revisiting samples from previously learned concepts. The authors demonstrate that this problem can be efficiently addressed using the concept of analytical learning. Initially, the paper received mixed feedback: 1x3 (reject), 1x4 (borderline reject), 1x6 (weak accept), and 1x8 (strong accept). After the rebuttal and the author-reviewer discussion period, the scores improved to 1x5 (borderline accept), 1x6 (weak accept), 1x7 (accept), and 1x8 (strong accept), with all reviewers recommending acceptance.

The reviewers acknowledged the novelty of the idea within the context of the problem, and the AC agrees with this assessment, recommending the paper for acceptance. Congratulations.

The AC suggests including all the results discussed during the rebuttal and the author-reviewer discussions. Additionally, the formulation of the classification problem as a regression, as done in the paper, can be acknowledged as a limitation of the current work.

Congratulations again.